# Score-based Neural Ordinary Differential Equations for Computing Mean Field Control Problems

## Abstract

Classical neural ordinary differential equations (ODEs) are powerful tools for approximating the log-density functions in high-dimensional spaces along trajectories, where neural networks parameterize the velocity fields. This paper proposes a system of neural differential equations representing first- and second-order score functions along trajectories based on deep neural networks. We reformulate the mean field control (MFC) problem with individual noises into an unconstrained optimization problem framed by the proposed neural ODE system. Additionally, we introduce a novel regularization term to enforce characteristics of viscous Hamilton–Jacobi–Bellman (HJB) equations to be satisfied based on the evolution of the second-order score function. Examples include regularized Wasserstein proximal operators (RWPOs), probability flow matching of Fokker–Planck (FP) equations, and linear quadratic (LQ) MFC problems, which demonstrate the effectiveness and accuracy of the proposed method.

## 1 Introduction

Score functions have been widely used in modern machine learning algorithms, particularly generative models through time-reversible diffusion (Song et al., 2021). The score function can be viewed as a deterministic representation of diffusion in stochastic trajectories (Carrillo et al., 2019). In this representation, one reformulates the Brownian motion by the gradient of the logarithm of the density function, after which deterministic trajectories involving score functions can approximate the probability density function. These properties have inspired algorithms for simulating stochastic trajectories or sampling problems that converge to target distributions (Wang et al., 2022; Lu et al., 2024). Typical applications include modeling the time evolution of probability densities for stochastic dynamics and solving control problems constrained by such dynamics.

While score functions provide powerful tools for modeling stochastic trajectories, their computations are often inefficient, especially in high-dimensional spaces. Classical methods, such as kernel density estimation (KDE) (Chen, 2017), tend to perform poorly in such settings due to the curse of dimensionality (Terrell & Scott, 1992).

Recently, neural ODEs have emerged as efficient ways of estimating densities. In particular, one uses neural networks to parameterize the velocity fields and then approximates the logarithm of density function along trajectories. The time discretizations of neural ODEs can be viewed as normalization flows in generative models.

Several natural questions arise. *Can we approximate the trajectory of score functions by constructing a set of neural ODEs (in continuous time) or normalization flows (in discrete time)? Furthermore, can we efficiently solve stochastic control problems with these neural ODEs or normalization flows?*

In this paper, we propose a formulation for the first- and second-order score functions using two additional neural ODEs involving high-order derivatives of the velocity fields. We also develop a class of high-order normalizing flows in the discrete-time update of these neural ODEs. As an application, we use this high-order normalizing flow to solve the MFC problem with individual noises. The MFC problem generalizes stochastic control problems by incorporating a running cost that depends on the state distribution, thus capturing the interaction between individual agents and the population

density. In our reformulation, the MFC problem is recast into an optimal control problem involving state trajectories, densities, and score functions, which can be efficiently approximated using the proposed neural ODE system. Additionally, a regularization term from the Karush-Kuhn-Tucker (KKT) system of the MFC problem is proposed to enhance the optimization. In this regularization, we approximate the viscous HJB equations using the first- and second-order score dynamics. Numerical examples in various MFC problems demonstrate the effectiveness of the proposed optimization method with high-order neural differential equations.

**Related work.** The normalizing flow has emerged as a powerful technique for solving inference problems, allowing for the construction of complex probability distributions in a tractable manner. This approach was popularized by works such as Rezende & Mohamed (2015); Papamakarios et al. (2021). The advent of Neural ODEs (Chen et al., 2018) has attracted considerable attention in the community, opening up a new paradigm for continuous-time deep learning models. Following this breakthrough, numerous extensions have been proposed to enhance the expressiveness and flexibility of these models, such as the augmented Neural ODEs by Dupont et al. (2019).

Regarding score-based models, Song et al. (2021) studies the score-based reversible time diffusion models in generative modeling. In this case, the score function often comes from the OU process. One needs to solve the score-matching problem to learn the time-dependent score function. In addition, the first-order score dynamic from neural ODEs has been introduced by Boffi & Vanden-Eijnden (2023) to work on the simulation of the Fokker-Planck (FP) equation. Rather than using score dynamics directly to compute the probability flow ODEs associated with FP equations, they employed score-matching methods to achieve efficient computations.

The study of MFC problems has become crucial in the last decade Cardaliaguet et al. (2019); Bensoussan et al. (2013); Fornasier & Solombrino (2014). MFC studies strategic decision-making in large populations where individual players interact through specific mean-field quantities. This control formulation is also useful in generative models Zhang & Katsoulakis (2023). For example, neural network-based methods have been employed to solve MFC problems. Hu & Lauriere (2023) provided a comprehensive review of neural network approaches for MFC. For first-order MFC problems, Ruthotto et al. (2020); Huang et al. (2023) developed Neural ODE-based methods. In parallel, Ruthotto et al. (2020) introduced numerical algorithms based on the characteristic lines of the Hamilton-Jacobi-Bellman (HJB) equation. For second-order MFC problems, neural networks have also been employed. Lin et al. (2021) utilized generative adversarial networks (GANs) to approximate minimization systems for MFC problems, where one neural network models the population density, and the other parameterizes the value function. Reisinger et al. (2024) proposed a PDE-based iterative algorithm to tackle MFC problems. Along this line, MFC problems are generalizations of dynamical optimal transport problems, known as Benamou-Brenier's formulas Benamou & Brenier (2000). A famous example is the Jordan-Kinderlehrer-Otto (JKO) scheme (Jordan et al., 1998), a variational time discretization in approximating the gradient drift FP equation. The one-step iteration of the JKO scheme can be viewed as an MFC problem. In machine learning computations, the neural JKO scheme (Xu et al., 2023; Vidal et al., 2023) was introduced to compute the FP equation, and Lee et al. (2024) extended this to approximate general nonlinear gradient flows through a generalized deep neural JKO scheme. This scheme approximates a deterministic MFC problem in each time interval. Other works that use machine learning methods to solve MFC problems include Dayanikli et al. (2023); Zhou et al. (2024); Dayanikli et al. (2023). Different from previous results, our work designs first- and second-order score dynamics to compute second-order MFC problems.

The organization of this paper is as follows. In section 2, we formulate the neural dynamical system involving first- and second-order score functions, with a specific example based on the Gaussian distribution in terms of linear mapping equations. Section 3 discusses the discrete time evolution of the neural ODE system, which can be viewed as a system of normalization flows for approximating first- and second-order score functions. In section 4, we design an algorithm using a score-based ODE system to solve the second-order MFC problem. Several numerical examples are presented in section 5 to demonstrate the accuracy and effectiveness of the proposed algorithm.

## 2 Score-based neural ODE in continuous time

In this section, we propose a neural dynamical system that evolves along a trajectory of a random variable. The system outputs the logarithm of the density function, as well as its gradient vector and

Hessian matrix. We show that these neural dynamics are crucial for efficiently computing the score functions in high-dimensional density estimation problems, which are key in generative models and MFC problems.

We first clarify the notations. Denote a variable $x \in \mathbb{R}^d$. $\nabla_x$ denotes the gradient or Jacobian matrix of a function w.r.t. the variable $x$. The gradient is always a column vector. $\nabla_x \cdot$ and $\nabla_x^2$ are divergence and Hessian w.r.t. the variable $x$. $|\cdot|$ is the absolute value of a scalar, the $l_2$ norm of a vector, or the Frobenius norm of a matrix. $\mathrm{Tr}(\cdot)$ denotes the trace of a squared matrix.

Let $z_t \in \mathbb{R}^d$ be a flow of random variable for $t \geq 0$ with initialization $z_0 = x$, which is obtained from a pushforward map $z_t = T(t, x)$ in Lagrangian coordinates. For simplicity, we denote $T_t := T(t, \cdot)$, so that $T_0$ is an identity map. Throughout this paper, we assume that $T(\cdot, \cdot)$ is smooth and $T_t$ is invertible for all $t \geq 0$. We denote the probability distribution of $z_t$ by $\rho_t$ or $\rho(t, \cdot)$, then $\rho_t = T_{t\#}\rho_0$, where $\#$ is the pushforward operator for distributions. By the change of variable formula, $\rho$ satisfies the well known Monge–Ampère (MA) equation (Gutiérrez & Brezis, 2001)

$$\rho(t, T(t, x)) \, \det(\nabla_x T(t, x)) = \rho(0, x) \,. \tag{1}$$

Let $f(t, \cdot) : \mathbb{R}^d \rightarrow \mathbb{R}^d$ be the vector field of the state dynamic $z_t$ in Eulerian coordinates, given by $f(t, T(t, x)) = \partial_t T(t, x)$ for all $t$ and $x$. Then, the state $z_t$ satisfies the ODE dynamic $\partial_t z_t = f(t, z_t)$. The density function hence satisfies the continuity equation (also known as the transport equation):

$$\partial_t \rho(t, x) + \nabla_x \cdot (\rho(t, x) f(t, x)) = 0 \,. \tag{2}$$

We leave the proofs for this equation and all propositions afterwards in Appendix A.

We denote the logarithm of density along the state trajectory $z_t$ by $l_t := \log \rho(t, z_t)$. We also denote the first- and second-order score functions along the state trajectory by $s_t := \nabla_z \log \rho(t, z_t)$ and $H_t := \nabla_z^2 \log \rho(t, z_t)$. These score functions are useful in density estimation problems, particularly for applications such as generative models and MFC problems. However, their efficient computations in high-dimensional spaces are challenging problems. In this work, we propose a system of high-order neural ODEs to compute these score functions, formalized in the following proposition.

**Proposition 1** (Neural ODE system). *Functions $z_t$, $l_t$, $s_t$, and $H_t$ satisfy the following ODE dynamics.*

$$\partial_t z_t = f(t, z_t) \,, \tag{3a}$$

$$\partial_t l_t = -\nabla_z \cdot f(t, z_t) \,, \tag{3b}$$

$$\partial_t s_t = -\nabla_z f(t, z_t)^\top s_t - \nabla_z (\nabla_z \cdot f(t, z_t)) \,, \tag{3c}$$

$$\partial_t H_t = -\sum_{i=1}^{d} s_{it} \nabla_z^2 f_i(t, z_t) - \nabla_z^2 (\nabla_z \cdot f(t, z_t)) - H_t \nabla_z f(t, z_t) - \nabla_z f(t, z_t)^\top H_t \,, \tag{3d}$$

*where $s_{it}$ and $f_i$ are the $i$-th component of $s_t$ and $f$ respectively.*

One corollary of this proposition is that, if we denote the density along the trajectory by $\tilde{l}_t := \rho(t, z_t)$, then $\tilde{l}_t$ satisfies the following ODE:

$$\partial_t \tilde{l}_t = -\nabla_z \cdot f(t, z_t) \, \tilde{l}_t \,. \tag{4}$$

We note that the first and second order score functions satisfy the following information equality.

**Proposition 2** (Information equality). *The following equality holds for all $t \geq 0$,*

$$\mathbb{E}\left[\mathrm{Tr}(H_t)\right] = -\mathbb{E}\left[|s_t|^2\right] \,.$$

**Example: Centered Gaussian distributions.** To illustrate the ODE dynamics (3), we consider a concrete example where the random variable follows a centered Gaussian distribution. Let the pushforward map be linear in $x$, i.e. $T(t, x) = T(t)x$, where $T(t) : \mathbb{R}_+ \rightarrow \mathbb{R}^{d \times d}$ is a time-dependent matrix. If we define $A(t) : \mathbb{R}_+ \rightarrow \mathbb{R}^{d \times d}$ by $\partial_t T(t) = A(t)T(t)$, then the vector field of the state dynamics is $f(t, x) = A(t)x$. Now, let the initial state $z_0$ follow a centered Gaussian distribution $N(0, \Sigma(0))$ with covariance matrix $\Sigma(0)$. Under this linear map $T(t)$, the state $z_t$ remains a Gaussian distribution, with a time-dependent covariance matrix $\Sigma(t)$. In this case, $z_t \sim N(0, \Sigma(t))$, and

the time evolution of the covariance matrix follows the matrix ODE (with the derivation in Appendix A.1)

$$\partial_t \Sigma(t) = A(t)\Sigma(t) + \Sigma(t)A(t)^\top .$$

In this case, $\nabla_x f(t,x) = A(t)$ and $\nabla_x \cdot f(t,x) = \text{Tr}(A(t))$, and all higher-order spatial derivatives of $f$ vanish. Therefore, the ODE system (3) simplifies to

$$\partial_t z_t = A(t)z_t , \qquad \partial_t l_t = -\text{Tr}(A(t)) ,$$
$$\partial_t s_t = -A(t)^\top s_t , \qquad \partial_t H_t = -H_t A(t) - A(t)^\top H_t .$$

# 3 SCORE-BASED NORMALIZATION FLOWS

In this section, we introduce the time discretization of the proposed neural ODE system. This discretized system can be interpreted as a normalization flow. This means that we can efficiently estimate the first- and second-order score functions using a deep neural network function.

## 3.1 A GENERALIZATION OF NORMALIZING FLOW

Chen et al. (2018) interpreted the deep residual neural network as an ODE with each layer representing one step of the forward Euler scheme. Similar ideas have also been proposed by Haber & Ruthotto (2017). In this section, we extend these concepts and demonstrate how the first- and second-order score functions can be computed efficiently using the proposed neural ODE system.

We partition the time interval $[0, t_{\text{end}}]$ into $N_t$ subintervals with the length $\Delta t = t_{\text{end}}/N_t$ and denote the time stamps by $t_j = j\,\Delta t$. We apply the forward Euler scheme to discretize the ODE system (3). We parametrize the vector field $f(t, z; \theta)$ as a neural network with parameter $\theta$. Here, $\theta = [w_0, w_1, w_2, b_1, b_2] \in \mathbb{R}^{k+k\times d+d\times k+k+d}$ and

$$f(t, z; \theta) = w_2\,\sigma(w_0 t + w_1 z + b_1) + b_2 , \tag{5}$$

where $\sigma\colon \mathbb{R}^k \to \mathbb{R}^k$ is a vector function with elementwise activation functions. Here $k$ is the width of the network. This is the typical structure of a neural network with one hidden layer. Then, we discretize the state dynamic $z_t$ through

$$z_{t_{j+1}} = z_{t_j} + \Delta t\, f(t_j, z_{t_j}; \theta) . \tag{6}$$

Similarly, numerical simulations for functions $l_t$, $\tilde{l}_t$, $s_t$, and $H_t$ are given by

$$l_{t_{j+1}} = l_{t_j} - \Delta t\, \nabla_z \cdot f(t_j, z_{t_j}; \theta) , \tag{7}$$

$$\tilde{l}_{t_{j+1}} = \tilde{l}_{t_j} - \Delta t\, \nabla_z \cdot f(t_j, z_{t_j}; \theta)\, \tilde{l}_{t_j} , \tag{8}$$

$$s_{t_{j+1}} = s_{t_j} - \Delta t\, \left( \nabla_z f(t_j, z_{t_j}; \theta)^\top s_{t_j} + \nabla_z(\nabla_z \cdot f(t_j, z_{t_j}; \theta)) \right) , \tag{9}$$

$$H_{t_{j+1}} = H_{t_j} - \Delta t\, \left( \sum_{i=1}^{d} s_{it_j} \nabla_z^2 f_i(t_j, z_{t_j}; \theta) + \nabla_z^2(\nabla_z \cdot f(t_j, z_{t_j}; \theta)) \right. \tag{10}$$

$$\left. + H_{t_j} \nabla_z f(t_j, z_{t_j}; \theta) + \nabla_z f(t_j, z_{t_j}; \theta)^\top H_{t_j} \right) ,$$

where the derivatives of $f$ are obtained from auto-differentiations of the neural network.

In this way, the map from $z_0, l_0, \tilde{l}_0, s_0, H_0$ to $z_{t_j}, l_{t_j}, \tilde{l}_{t_j}, s_{t_j}, H_{t_j}$, for any $j \geq 2$, can be viewed as deep residual neural networks, with each layer given by a forward Euler time step. For example,

$$z_{t_{N_t}} = (id + \Delta t\, f(t_{N_t-1}, \cdot; \theta)) \circ \cdots \circ (id + \Delta t\, f(t_0, \cdot; \theta))\, (z_0) ,$$
$$l_{t_{N_t}} = \left(id - \Delta t\, \nabla_z \cdot f(t_{N_t-1}, z_{t_{N_t-1}}; \theta)\right) \circ \cdots \circ \left(id - \Delta t\, \nabla_z \cdot f(t_0, z_0; \theta)\right)\, (l_0) ,$$
$$s_{t_{N_t}} = \left(id - \Delta t\, \left(\nabla_z f(t_{N_t-1}, z_{N_t-1}; \theta)^\top \cdot + \nabla_z(\nabla_z \cdot f(t_{N_t-1}, z_{N_t-1}; \theta))\right)\right) \circ \cdots \circ \tag{11}$$
$$\left(id - \Delta t\, \left(\nabla_z f(t_0, z_0; \theta)^\top \cdot + \nabla_z(\nabla_z \cdot f(t_0, z_0; \theta))\right)\right)\, (s_0) ,$$

where $id$ is the identity mapping function and $\circ$ represents compositions of functions.

## 3.2 DIFFERENT ALGORITHMS FOR COMPUTING THE SCORE FUNCTION

The expression (3c) or (9) provides a way to compute the score function. In this section, we compare several algorithms for computing the score function, and demonstrate the potential efficiency of using formulas (3c) or (9).

One method is to derive the score function from the **pushforward map** $T(t, x)$ and the MA equation directly. To be more specific, we apply the operator $\nabla_x$ and the logarithm on both sides of the MA equation (1). We obtain

$$\nabla_x T(t,x)^\top \nabla_T \log \rho(t, T(t,x)) + \nabla_x \log\left(\det(\nabla_x T(t,x))\right) = \nabla_x \log \rho(0, x) .$$

Recall that $s_t = \nabla_T \log \rho(t, T(t,x))$, so the score function can be computed through

$$s_t = \nabla_x T(t,x)^{-\top} \left[ \nabla_x \log \rho(0, x) - \nabla_x \log\left(\det(\nabla_x T(t,x))\right) \right] . \tag{12}$$

This formulation requires computing the inverse of Jacobian or solving related linear systems, resulting in a cost of $\mathcal{O}(d^3)$ for a single $s_t$. As a consequence, the total cost for computing one score trajectory $\{s_{t_j}\}_{j=0}^{N_t}$ is $\mathcal{O}(N_t d^3)$.

As an alternative, we can compute the score $s_t$ through the **normalizing flow** $l_t$, i.e., differentiating $l_t = \log \rho(t, z_t)$ w.r.t $z_t$. We assume that the width of the neural network (5) is $k = \mathcal{O}(d)$. The score function can be reformulated as

$$s_t = \nabla_T \log \rho(t, T(t,x)) = \nabla_x T(t,x)^{-\top} \nabla_x \log \rho(t, T(t,x)) = \left(\frac{\partial z_t}{\partial z_0}\right)^{-\top} \frac{\partial l_t}{\partial z_0} , \tag{13}$$

where $\dfrac{\partial z_t}{\partial z_0}$ and $\dfrac{\partial l_t}{\partial z_0}$ are obtained from auto-differentiations of deep neural network functions in (11). Using the chain rule, the total computational computational cost for computing one trajectory of the score function is still $\mathcal{O}(N_t d^3)$. Both (12) and (13) require computing the inverse Jacobian of $T$ or solving related linear systems, resulting in a cubic cost in the spatial dimension. As mentioned by Chen et al. (2018) (section 4), this is the bottleneck of these methods. We leave more detailed discussions of both methods (12) and (13), and two other methods, to Appendix D.

In contrast, the **high-order normalizing flow** (9) only requires discretizing the ODE dynamics (3a) and (3c), with a total cost of $\mathcal{O}(N_t d^2)$ for computing one score trajectory $\{s_{t_j}\}_{j=0}^{N_t}$. As is shown in table 1, our formulation is more efficient compared with (12) and (13), especially in high dimensions. Additionally, we are able to compute the second-order score function through 10.

| pushforward map (12) | normalizing flow (13) | high-order normalizing flow (9) (ours) |
|:---:|:---:|:---:|
| $\mathcal{O}(N_t d^3)$ | $\mathcal{O}(N_t d^3)$ | $\mathcal{O}(N_t d^2)$ |

Table 1: Complexity for different algorithms to compute a score trajectory.

**Example: Gaussian distribution.** Our formulation for the second-order score function also has advantages in the example of Gaussian distributions, where $H_t = \Sigma(t)^{-1}$ is exactly the inverse of the covariance matrix. Traditionally, estimating $\Sigma(t)^{-1}$ involves several steps. One needs to sample sufficiently many points $z_0^{(n)}$, simulate the state dynamic to obtain $z_t^{(n)}$, estimate the covariance matrix from these samples, and finally compute the inverse of this estimation. To achieve an error of $\varepsilon$, at least $\mathcal{O}(\varepsilon^{-2})$ samples are required for accurate covariance estimation. Compared to our second-order score dynamic (3d), we only need to pick $\Delta t = \mathcal{O}(\varepsilon)$ and compute $H_t$ through (10), with a total cost of $\mathcal{O}(\varepsilon^{-1})$. This results in a significantly more efficient algorithm.

## 4 SOLVING SECOND-ORDER MFC PROBLEMS

In this section, we apply the score-based normalizing flow to solve the second-order MFC problem. In particular, we demonstrate that the MFC problem can be represented by the optimal control problem of the neural dynamical system in Proposition 1. For readers interested in the connection between MFC problems and generative AI, we refer to the work of Zhang & Katsoulakis (2023).

In the context of machine learning, Neklyudov et al. (2023) also studies similar problems, in which they compute Wasserstein Lagrangian flows. In this work, we can deal with the diffusion term in MFC problems using proposed neural normalization flows.

## 4.1 Formulation of MFC problem

We consider the following MFC problem

$$\inf_v \int_0^{t_{\text{end}}} \int_{\mathbb{R}^d} \left[ L(t, x, v(t, x)) + F(t, x, \rho(t, x)) \right] \rho(t, x)\, \mathrm{d}x\, \mathrm{d}t + \int_{\mathbb{R}^d} G(x, \rho(t_{\text{end}}, x))\rho(t_{\text{end}}, x)\, \mathrm{d}x \,, \tag{14}$$

where the density $\rho(t, x)$ satisfies the FP equation with a given initialization

$$\partial_t \rho(t, x) + \nabla_x \cdot (\rho(t, x)v(t, x)) = \gamma \Delta_x \rho(t, x) \,, \quad \rho(0, x) = \rho_0(x) \,. \tag{15}$$

The corresponding stochastic dynamic is

$$\mathrm{d}X_t = v(t, X_t)\, \mathrm{d}t + \sqrt{2\gamma}\, \mathrm{d}W_t \,, \qquad X_0 \sim \rho_0 \,.$$

Here $L : \mathbb{R} \times \mathbb{R}^d \times \mathbb{R}^d \to \mathbb{R}$ is the running cost function, and we assume that it is strongly convex in $v$. $F : \mathbb{R} \times \mathbb{R}^d \times \mathbb{R} \to \mathbb{R}$ is the cost that involves the density, which distinguishes the MFC problem from the optimal control. $G : \mathbb{R}^d \times \mathbb{R} \to \mathbb{R}$ is the terminal cost, which may also involve the density.

Given the density function $\rho(\cdot, \cdot)$, we define the composed velocity field $f$ as

$$f(t, x) = v(t, x) - \gamma \nabla_x \log \rho(t, x) \,. \tag{16}$$

Let $z_0 = x_0$, under the vector field $f$, we obtain the probability flow (Chen et al., 2024) of the stochastic dynamic $x_t$, given by $\partial_t z_t = f(t, z_t)$, which coincides with (3a). This deterministic dynamic characterizes the probability distribution $\rho(t, x)$. If $z_0 \sim \rho_0$, then the probability distribution for $z_t$ is exactly $\rho(t, \cdot)$. With this transformation, the velocity $v(t, x)$ becomes $f(t, x) + \gamma \nabla_x \log \rho(t, x)$, and the MFC problem follows

$$\inf_f \int_0^{t_{\text{end}}} \int_{\mathbb{R}^d} \left[ L\left(t, x, f(t, x) + \gamma \nabla_x \log \rho(t, x)\right) + F(t, x, \rho(t, x)) \right] \rho(t, x)\, \mathrm{d}x\, \mathrm{d}t$$
$$+ \int_{\mathbb{R}^d} G(x, \rho(t_{\text{end}}, x))\rho(t_{\text{end}}, x)\, \mathrm{d}x \,, \tag{17}$$

subject to the transport equation

$$\partial_t \rho(t, x) + \nabla_x \cdot (\rho(t, x)f(t, x)) = 0 \,, \quad \rho(0, x) = \rho_0(x) \,. \tag{18}$$

## 4.2 Modified HJB equation for MFC

In this section, we present a system of two PDEs that characterize the optimal solution for the MFC problem. Different from the traditional FP-HJB pair (see (Bensoussan et al., 2013, Chapter 4)), our system consists of a transport equation obtained from (18), and a modified HJB equation, tailored for the modified MFC problem (17). This characterization could serve as a regularizer to enhance the loss function in numerical algorithms. We define the Hamiltonian $H : \mathbb{R} \times \mathbb{R}^d \times \mathbb{R}^d \to \mathbb{R}$ by

$$H(t, x, p) = \sup_{v \in \mathbb{R}^d} \left( v^\top p - L(t, x, v) \right) \,.$$

This definition aligns with classical control theory and is closely related to the maximum principle (Zhou & Lu, 2023). The solution of the MFC problem is summarized by the following proposition.

**Proposition 3.** *Let $L$ be strongly convex in $v$, then the solution to the MFC problem* (17) *is as follows. Consider a function $\psi \colon [0, t_{end}] \times \mathbb{R}^d \to \mathbb{R}$, such that*

$$f(t, x) = \mathrm{D}_p H(t, x, \nabla_x \psi(t, x) + \gamma \nabla_x \log \rho(t, x)) - \gamma \nabla_x \log \rho(t, x) \,, \tag{19}$$

*where the density function $\rho(t, x)$ and $\psi \colon [0, t_{end}] \times \mathbb{R}^d \to \mathbb{R}$ satisfy the following system of equations*

$$\begin{cases} \partial_t \rho(t, x) + \nabla_x \cdot (\rho(t, x)\mathrm{D}_p H) = \gamma \Delta_x \rho(t, x) \,, \\ \partial_t \psi(t, x) + \nabla_x \psi(t, x)^\top \mathrm{D}_p H - \gamma \nabla_x \cdot \mathrm{D}_p H + \gamma \Delta_x \psi(t, x) - L(t, x, \mathrm{D}_p H) \\ \quad - \widetilde{F}(t, x, \rho(t, x)) + 2\gamma^2 \Delta_x \log \rho(t, x) + \gamma^2 \left| \nabla_x \log \rho(t, x) \right|^2 = 0 \,, \\ \rho(0, x) = \rho_0(x) \,, \quad \psi(t_{end}, x) = -\widetilde{G}(x, \rho(t_{end}, x)) - \gamma \log \rho(t_{end}, x) \,, \end{cases} \tag{20}$$

*Here, $D_p H$ is short for $D_p H(t, x, \nabla_x \psi(t, x) + \gamma \nabla_x \log \rho(t, x))$, $\widetilde{F}(t, x, \rho) = \frac{\partial}{\partial \rho}(F(t, x, \rho)\rho) =$*

$\frac{\partial F}{\partial \rho}(t, x, \rho)\rho + F(t, x, \rho)$, *and* $\widetilde{G}(x, \rho) = \frac{\partial G}{\partial \rho}(x, \rho)\rho + G(x, \rho)$.

In the LQ problem, we let $L(t, x, v) = \frac{1}{2}|v|^2$ and $F(t, x, \rho) = 0$. The MFC problem becomes

$$\inf_f \int_0^{t_{\text{end}}} \int_{\mathbb{R}^d} \frac{1}{2}\left|f(t, x) + \gamma \nabla_x \log \rho(t, x)\right|^2 \rho(t, x)\, dx\, dt + \int_{\mathbb{R}^d} G(x, \rho(t_{\text{end}}, x))\rho(t_{\text{end}}, x)\, dx\,, \quad (21)$$

subject to (18). In this case, the result becomes the Corollary below.

**Corollary 1.** *In the LQ problem where $L(t, x, v) = \frac{1}{2}|v|^2$ and $F(t, x, \rho) = 0$, the solution of the MFC problem* (21) *is given by*

$$f(t, x) = \nabla_x \psi(t, x)\,,$$

*and*

$$\begin{cases} \partial_t \rho(t, x) + \nabla_x \cdot (\rho(t, x)\nabla_x \psi(t, x)) = 0\,, \\ \partial_t \psi(t, x) + \frac{1}{2}\left|\nabla_x \psi(t, x)\right|^2 + \gamma^2 \Delta_x \log \rho(t, x) + \frac{1}{2}\gamma^2 \left|\nabla_x \log \rho(t, x)\right|^2 = 0\,, \quad (22) \\ \rho(0, x) = \rho_0(x)\,, \quad \psi(t_{end}, x) = -\widetilde{G}(x, \rho(t_{end}, x)) - \gamma \log \rho(t_{end}, x)\,. \end{cases}$$

*If we further define the Fisher information as $I[\rho] := \int_{\mathbb{R}^d} \left|\nabla_x \log \rho(x)\right|^2 \rho(x)\, dx$, then the equation for $\psi$ in equation* (22) *becomes*

$$\partial_t \psi(t, x) + \frac{1}{2}\left|\nabla_x \psi(t, x)\right|^2 - \frac{1}{2}\gamma^2 \frac{\delta I[\rho(t, \cdot)]}{\delta \rho(t, \cdot)}(x) = 0\,.$$

Let $z_t$ be the characteristic line of the state trajectory, satisfying $\partial_t z_t = f(t, z_t)$ where $f$ is given by (19). Then, a residual of the HJB equation along the trajectory $z_t$ can be computed to enhance the objective function. For example, in the LQ example in Corollary 1, we know that $\psi$ satisfies

$$\partial_t \psi(t, z_t) + \frac{1}{2}\left|\nabla_z \psi(t, z_t)\right|^2 + \gamma^2 \left(\text{Tr}(H_t) + \frac{1}{2}|s_t|^2\right) = 0\,. \quad (23)$$

Also, by (16), we can recover the optimal velocity field $v$ through

$$v(t, z_t) = f(t, z_t) + \gamma s_t\,,$$

where $f$ is the optimal vector field of the modified MFC problem. This regularization technique could be extended to a more general setting, such as the flow matching problem for overdamped Langevin dynamics. See Corollary 2 in Appendix A for details.

### 4.3 NUMERICAL ALGORITHMS

In this section, we present numerical algorithms to solve the MFC problem. We minimize the objective (17) to obtain the optimal vector field $f$. Additionally, we can incorporate the residual of the HJB equation as a regularizer to enhance the loss functional, as discussed after Corollary 1.

The composed velocity field $f$ is parametrized as a neural network, as defined in (5). In order to simulate the cost functional numerically, we sample multiple initial points $z_0^{(n)} \sim \rho_0$ with a batch size $N_z$. Then, we can simulate the dynamics of $z_t^{(n)}, l_t^{(n)}, \tilde{l}_t^{(n)}, s_t^{(n)}$, and $H_t^{(n)}$ numerically for each particle through (6), (7), (8), (9), and (10) respectively, where the derivatives of $f$ are obtained via auto-differentiation. Then, we simulate the cost functional (17) by

$$\mathcal{L}_{\text{cost}} = \frac{1}{N_z} \sum_{n=1}^{N_z} \left[ \sum_{j=0}^{N_t-1} \left( L\left(t_j, z_{t_j}^{(n)}, f(t_j, z_{t_j}^{(n)}; \theta) - \gamma s_{t_j}^{(n)}\right) + F\left(t_j, z_{t_j}^{(n)}, \tilde{l}_{t_j}^{(n)}\right)\right) \Delta t + G(z_{t_{N_t}}^{(n)}, \tilde{l}_{t_{N_t}}^{(n)}) \right]\,.$$

$$(24)$$

With this simulation cost, we minimize this loss $\mathcal{L}_{\text{cost}}$ using the Adam method. The algorithm is summarized in Algorithm 1.

---

**Algorithm 1** Score-based normalizing flow solver for the MFC problem

---

**Input:** MFC problem (14) (15), $N_t$, $N_z$, network structure (5), learning rate, number of iterations
**Output:** the solution to the MFC problem
   Initialize $\theta$
   **for** index $= 1$ **to** index$_{\text{end}}$ **do**
      Sample $N_z$ points $\{z_0^{(n)}\}_{n=1}^{N_z}$ from the initial distribution $\rho_0$
      Compute $\tilde{l}_0^{(n)} = \rho(0, z_0^{(n)})$, $s_0^{(n)} = \nabla_z \log \rho(0, z_0^{(n)})$
      Initialize loss $\mathcal{L}_{\text{cost}} = 0$
      **for** $j = 0$ **to** $N_t - 1$ **do**

$$\text{update loss } \mathcal{L}_{\text{cost}} \mathrel{+}= \frac{1}{N_z} \sum_{n=1}^{N_z} \left( L(t_j, z_{t_j}^{(n)}, f(t_j, z_{t_j}^{(n)}; \theta) - \gamma s_{t_j}^{(n)}) + F(t_j, z_{t_j}^{(n)}, \tilde{l}_{t_j}^{(n)}) \right) \Delta t$$

         compute $(\nabla_z, \nabla_z \cdot, \nabla_z(\nabla_z \cdot)) f(t_j, z_{t_j}^{(n)}; \theta)$ according to (5)
         compute $z_{t_{j+1}}^{(n)}, \tilde{l}_{t_{j+1}}^{(n)}, s_{t_{j+1}}^{(n)}$ through the forward Euler scheme (6), (8), and (9)
      **end for**

$$\text{add terminal cost } \mathcal{L}_{\text{cost}} \mathrel{+}= \frac{1}{N_z} \sum_{n=1}^{N_z} G(z_{t_{N_t}}^{(n)}, \tilde{l}_{t_{N_t}}^{(n)})$$

      update the parameters $\theta$ through Adam method to minimize the loss $\mathcal{L}_{\text{cost}}$
   **end for**

---

To further improve performance, we can add a penalty of the residual of the HJB equation, denoted by $\mathcal{L}_{\text{HJB}}$, to enhance the loss function. This regularization technique is detailed in Appendix B.1. The total loss function is hence given by

$$\mathcal{L}_{\text{total}} = \mathcal{L}_{\text{cost}} + \lambda \mathcal{L}_{\text{HJB}} \,, \tag{25}$$

where $\lambda \geq 0$ is a weight parameter.

## 5 NUMERICAL RESULTS

We present numerical results for several examples in this section, including the regularized Wasserstein proximal operator (RWPO), the flow matching problem, a linear-quadratic (LQ) problem with an entropy potential cost, and an example with double well potential. For examples with exact solutions, we present all errors for the density (err$_\rho$), velocity field (err$_f$), and score function (err$_s$). These errors are calculated as averages over 10 independent runs, see (52) in Appendix B.3 for details. The parameters used for all numerical examples are provided in Appendix B.4. Part of the results are deferred to Appendix C.

### 5.1 REGULARIZED WASSERSTEIN PROXIMAL OPERATOR

In the RWPO problem, the objective is a regularized Benamou-Brenier formulation for optimal transportation (Benamou & Brenier, 2000). We minimize the cost functional

$$\inf_v \int_0^1 \int_{\mathbb{R}^d} \frac{1}{2} \left| v(t,x) \right|^2 \rho(t,x) \, dx \, dt + \int_{\mathbb{R}^d} G(x) \rho(1,x) \, dx \,,$$

subject to the FP equation $\partial_t \rho(t,x) + \nabla_x \cdot (\rho(t,x) v(t,x)) = \gamma \Delta_x \rho(t,x)$. Here we set $G(x) = |x|^2/2$ and $\rho_0(x) = (8\pi)^{-d/2} \exp\left(-|x|^2/8\right)$. We test Algorithm 1 on this problem in 1, 2, and 10 dimensions. These errors are summarized in Table 2. Detailed definitions for err$_\rho$, err$_f$, and err$_s$ are given in Appendix B.3. Additionally, we report the cost gap, which represents the difference between the computed cost and the optimal cost, averaged over 10 independent runs. The results are also visualized in Figure 1. The plot on the left shows the cost functional through training in 1 dimension, which becomes close to the optimal cost in red. The plot in the middle compares the evolution of the density function computed through (8) under trained velocity with the true density evolution. Our density dynamic (4) accurately captures the density evolution. The plot on the right shows the particle trajectories of $z_t$ in 2 dimensions and compares them with the stochastic dynamics. Our

| errors | $\text{err}_\rho$ | $\text{err}_f$ | $\text{err}_s$ | cost gap |
|---|---|---|---|---|
| $1d$ | $3.52 \times 10^{-3}$ | $3.45 \times 10^{-2}$ | $8.52 \times 10^{-3}$ | $1.33 \times 10^{-2}$ |
| $2d$ | $6.53 \times 10^{-3}$ | $3.40 \times 10^{-2}$ | $4.49 \times 10^{-2}$ | $1.24 \times 10^{-2}$ |
| $10d$ | $1.68 \times 10^{-3}$ | $3.15 \times 10^{-2}$ | $6.42 \times 10^{-2}$ | $1.69 \times 10^{-1}$ |

Table 2: Errors for the RWPO problem.

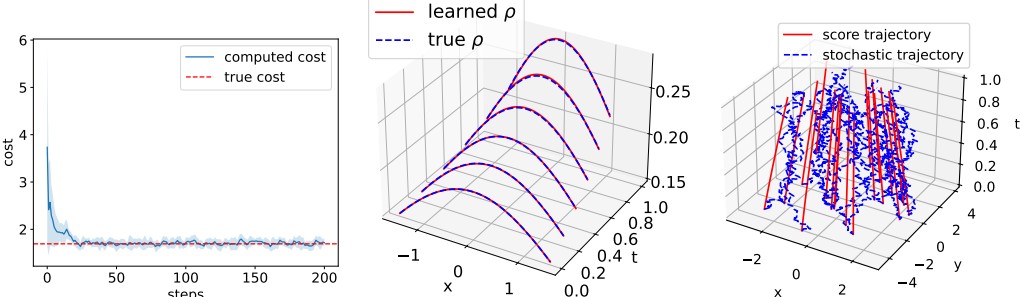

Figure 1: Numerical results for the RWPO problem. Left: cost functional through training with optimal cost in $1d$. Middle: density evolution through (4) and comparison with true density in $1d$. Right: trajectories of $z_t$ and comparison with the stochastic dynamic. The score dynamic demonstrates a structured behavior compared with the corresponding stochastic trajectory simulated by stochastic differential equations.

probability flow ODE demonstrates a structured behavior. We leave further numerical results of this example in Appendix C.1.

**Adding HJB regularizer.** We also test the regularized Algorithm 2 for this example. We compare all results with weight parameters $\lambda = 1 \times 10^{-3}$ and $\lambda = 0$ in (25) to study the effect of this regularization. All results are presented in table 3. The errors for the regularized algorithm are significantly smaller.

| regularization | $\lambda = 1 \times 10^{-3}$ | | $\lambda = 0$ | |
|---|---|---|---|---|
| errors | $\text{err}_A$ | $\text{err}_B$ | $\text{err}_A$ | $\text{err}_B$ |
| $1d$ | $2.33 \times 10^{-3}$ | $3.03 \times 10^{-3}$ | $1.36 \times 10^{-2}$ | $1.00 \times 10^{-2}$ |
| $2d$ | $3.30 \times 10^{-3}$ | $4.10 \times 10^{-3}$ | $1.74 \times 10^{-2}$ | $1.70 \times 10^{-2}$ |
| $10d$ | $7.52 \times 10^{-3}$ | $4.45 \times 10^{-3}$ | $8.34 \times 10^{-2}$ | $4.02 \times 10^{-2}$ |

Table 3: Errors for regularized MFC solver of RWPO problem. The errors for the regularized algorithm are significantly smaller.

## 5.2 FLOW MATCHING FOR SOLVING FP EQUATIONS

Flow matching has emerged as an important problem, which is closely related to generative models (Lipman et al., 2022; Boffi & Vanden-Eijnden, 2023). The problem is to simulate the probability density function of a stochastic dynamics:

$$\mathrm{d}X_t = b(t, X_t)\,\mathrm{d}t + \sqrt{2\gamma}\,\mathrm{d}W_t, \quad X_0 \sim \rho_0 \tag{26}$$

where $b\colon \mathbb{R}^+ \times \mathbb{R}^d \to \mathbb{R}^d$ is a known drift vector field, $\rho_0\colon \mathbb{R}^d \to \mathbb{R}$ is an initial value probability density function. In this example, we also assume the drift function satisfies $b(t, x) = -\nabla V(x)$, which is the negative gradient of some potential function $V(x) \in C_{\text{loc}}^4(\mathbb{R}^d)$. In this case, SDE (26) is the overdamped Langevin dynamic. To simulate the density function of stochastic process $X_t$ in (26), we design the following MFC problem. We minimize the objective functional (loss function)

as

$$\inf_v \int_0^{t_{\text{end}}} \int_{\mathbb{R}^d} \frac{1}{2} \left| v(t,x) - b(t,x) \right|^2 \rho(t,x) \, \mathrm{d}x \, \mathrm{d}t \,,$$

subject to the FP equation (15). After the score transformation (16), the above second order MFC problem becomes

$$\inf_f \int_0^{t_{\text{end}}} \int_{\mathbb{R}^d} \frac{1}{2} \left| f(t,x) - b(t,x) + \gamma \nabla_x \log \rho(t,x) \right|^2 \rho(t,x) \, \mathrm{d}x \, \mathrm{d}t \,,$$

subject to the transport equation (18). The state dynamic hence becomes $\partial_t z_t = -\nabla_z V(z_t) - \gamma s_t$. This transformation (16) seems to complicate the problem. However, we are able to learn the whole FP equation of the overdamped Langevin dynamic with a given initial distribution $\rho_0$ or samples $\{z_0^{(n)}\}_{n=1}^{n_z}$. More importantly, we can record all intermediate time steps and compute the density function during the entire time domain $[0, t_{\text{end}}]$.

We first present the flow matching problem for an Ornstein—Uhlenbeck (OU) process, where an explicit solution is available, given in Appendix A.3. The algorithm is described with details in Appendix B.1 and summarized in Algorithm 3. Similar to the previous section, we compare the numerical results with and without regularizations, presented in Table 4. The errors for the regularized algorithm are significantly smaller. We also provide visualized results in Figure 3 and 4 in Appendix C.2, which demonstrates the accuracy of our algorithm.

| regularization | $\lambda = 1 \times 10^{-3}$ | | $\lambda = 0$ | |
|---|---|---|---|---|
| errors | $\text{err}_A$ | $\text{err}_B$ | $\text{err}_A$ | $\text{err}_B$ |
| $1d$ | $9.81 \times 10^{-4}$ | $8.58 \times 10^{-4}$ | $7.89 \times 10^{-3}$ | $7.47 \times 10^{-3}$ |
| $2d$ | $1.50 \times 10^{-3}$ | $1.37 \times 10^{-3}$ | $2.02 \times 10^{-2}$ | $1.40 \times 10^{-2}$ |
| $10d$ | $4.08 \times 10^{-3}$ | $6.73 \times 10^{-3}$ | $7.86 \times 10^{-2}$ | $3.89 \times 10^{-2}$ |

Table 4: Errors for the regularized MFC solver of flow matching for OU processes. The errors for the regularized algorithm are significantly smaller.

In addition, we design an algorithm that partitions a long time interval into several sub-intervals, which could potentially resolve the issue of long time horizon. The model is trained consecutively over each sub-interval, referred to as the **multi-stage splicing method**. We present an example in 2 dimensions to demonstrate effectiveness of this splicing method, in which the invariant distribution of the SDE (26) is in a double moon shape (cf. (54)). Our splicing method (Algorithm 4) is able to capture the state dynamic within each interval, and pass the information to the next interval. The detailed numerical implementation and numerical results are presented in Appendix C.2.

We also present two more examples in Appendix C.3 and C.4, where we apply the proposed neural ODE system to approximate the LQ MFC problems.

## 6 CONCLUSION

In this paper, we propose a neural ODE system to compute evolutions of first- and second-order score functions along trajectories. The forward Euler discretization of neural ODE system satisfies a system of normalization flows along a deep neural network. We then apply the proposed neural ODE system to solve second-order MFC problems. The effectiveness and accuracy of our method are validated through numerical examples of RWPO problems, score-based flow matching problems for FP equations, LQ problems, and the double well potential problems, providing reassurance of its reliability and effectiveness.

In future work, we shall conduct a thorough numerical analysis of the optimal control problem within the neural ODE system. Specifically, for second-order MFC problems, error analysis for the score estimations is necessary. Although we observe some numerical advantages in using the score function to approximate the solution of viscous HJB equation, the underlying error in the current numerical scheme remains unclear. Additionally, exploring neural ODE systems for inverse problems presents an exciting avenue for future research. This direction involves learning and approximating stochastic trajectories from data using the neural ODE system.

## ETHICS STATEMENT

This work does not present any ethical concerns. It does not involve human subjects, personal data, or real-world deployment that could raise ethical issues such as privacy, safety, or fairness concerns.

## REPRODUCIBILITY STATEMENT

We provide the source code for all experiments as a supplementary material. The experiments were implemented using Python, based on generation of random variables, which can be reproduced directly.

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

# A   TECHNICAL DETAILS FOR SCORE-BASED NORMALIZING FLOWS AND MFC PROBLEMS

We present detailed proofs of all propositions about normalization flows and MFC problems in this section.

## A.1   PROOFS FOR SCORE-BASED NORMALIZING FLOW

**Continuity equation for density.** Let $T(t, x)$ be smooth and $T(t, \cdot)$ be invertible for all $t \geq 0$. Let $f(t, \cdot) : \mathbb{R}^d \to \mathbb{R}^d$ be the vector field of the state dynamic $z_t$, i.e., $f(t, T(t, x)) = \partial_t T(t, x)$ for all $t$ and $x$. Then, the probability density function satisfies the continuity equation

$$\partial_t \rho(t, x) + \nabla_x \cdot (\rho(t, x) f(t, x)) = 0 \,.$$

*Proof.* Using the chain rule, we have

$$\partial_t \nabla_x T(t, x) = \nabla_x \partial_t T(t, x) = \nabla_x f(t, T(t, x)) = \nabla_T f(t, T(t, x)) \, \nabla_x T(t, x) \,,$$

which implies

$$\mathrm{Tr} \left[ \partial_t \nabla_x T(t, x) \, \nabla_x T(t, x)^{-1} \right] = \mathrm{Tr} \left[ \nabla_T f(t, T(t, x)) \right] = \nabla_T \cdot f(t, T(t, x)) \,. \tag{27}$$

If we compute the derivative of the MA equation (1) w.r.t. $t$, we obtain

$$\begin{aligned}
0 &= \frac{\mathrm{d}}{\mathrm{d}t} \left[ \rho(t, T(t, x)) \det \left( \nabla_x T(t, x) \right) \right] \\
&= \frac{\mathrm{d}}{\mathrm{d}t} \rho(t, T(t, x)) \det \left( \nabla_x T(t, x) \right) + \rho(t, T(t, x)) \frac{\mathrm{d}}{\mathrm{d}t} \det \left( \nabla_x T(t, x) \right) \\
&= \left[ \partial_t \rho(t, T(t, x)) + \nabla_T \rho(t, T(t, x)) \, \partial_t T(t, x) \right] \det \left( \nabla_x T(t, x) \right) \\
&\quad + \rho(t, T(t, x)) \, \mathrm{Tr} \left[ \partial_t \nabla_x T(t, x) \, \nabla_x T(t, x)^{-1} \right] \det \left( \nabla_x T(t, x) \right) \\
&= \left[ \partial_t \rho(t, T(t, x)) + \nabla_T \rho(t, T(t, x))^\top f(t, T(t, x)) + \rho(t, T(t, x)) \nabla_T \cdot f(t, T(t, x)) \right] \\
&\quad \cdot \det \left( \nabla_x T(t, x) \right) \\
&= \left[ \partial_t \rho(t, T(t, x)) + \nabla_T \cdot (\rho(t, T(t, x)) f(t, T(t, x))) \right] \cdot \det \left( \nabla_x T(t, x) \right) \,,
\end{aligned}$$

where we have used (27) in the fourth equality. Since $T(t, \cdot)$ is invertible, $\det \left( \nabla_x T(t, x) \right) \neq 0$, which implies

$$\partial_t \rho(t, T(t, x)) + \nabla_T \cdot (\rho(t, T(t, x)) f(t, T(t, x))) = 0 \,,$$

for all $t$ and $x$. Therefore,

$$\partial_t \rho(t, x) + \nabla_x \cdot (\rho(t, x) f(t, x)) = 0 \,.$$

$\square$

**Proposition 1** (Neural ODE system). *Functions $z_t$, $l_t$, $s_t$, and $H_t$ satisfy the following ODE dynamics.*

$$\begin{aligned}
\partial_t z_t &= f(t, z_t) \,, \\
\partial_t l_t &= -\nabla_z \cdot f(t, z_t) \,, \\
\partial_t s_t &= -\nabla_z f(t, z_t)^\top s_t - \nabla_z (\nabla_z \cdot f(t, z_t)) \,, \\
\partial_t H_t &= -\sum_{i=1}^d s_{it} \nabla_z^2 f_i(t, z_t) - \nabla_z^2 (\nabla_z \cdot f(t, z_t)) - H_t \nabla_z f(t, z_t) - \nabla_z f(t, z_t)^\top H_t \,,
\end{aligned}$$

*where $s_{it}$ and $f_i$ are the $i$-th component of $s_t$ and $f$ respectively. Also, $\tilde{l}_t = \rho(t, z_t)$ satisfies*

$$\partial_t \tilde{l}_t = -\nabla_z \cdot f(t, z_t) \, \tilde{l}_t \,.$$

*Proof.* The dynamic for $z_t$ (3a) comes from the definition directly:

$$\partial_t z_t = \partial_t T(t, z_0) = f(t, T(t, z_0)) = f(t, z_t).$$

We next show (4) and (3b).

$$\partial_t \tilde{l}_t = \frac{\mathrm{d}}{\mathrm{d}t} \rho(t, z_t) = \partial_t \rho(t, z_t) + \nabla_z \rho(t, z_t)^\top \partial_t z_t = \partial_t \rho(t, z_t) + \nabla_z \rho(t, z_t)^\top f(t, z_t)$$

$$= -\nabla_z \cdot [f(t, z_t) \rho(t, z_t)] + \nabla_z \rho(t, z_t)^\top f(t, z_t) = -\nabla_z \cdot f(t, z_t) \rho(t, z_t) = -\nabla_z \cdot f(t, z_t) \tilde{l}_t.$$

where we have used the continuity equation (2) in the fourth equality. Consequently,

$$\partial_t l_t = \frac{\mathrm{d}}{\mathrm{d}t} \rho(t, z_t) / \rho(t, z_t) = -\nabla_z \cdot f(t, z_t) = -\nabla_z \cdot f(t, z_t).$$

We next show (3c). We will use the fact frequently that $\nabla_z \log \rho(t, z_t) = \nabla_z \rho(t, z_t) / \rho(t, z_t)$. We first compute $\frac{\mathrm{d}}{\mathrm{d}t} (\nabla_z \rho(t, z_t))$

$$\frac{\mathrm{d}}{\mathrm{d}t} [\nabla_z \rho(t, z_t)] = (\partial_t \nabla_z) \rho(t, z_t) + \nabla_z^2 \rho(t, z_t) \, \partial_t z_t = \nabla_z \partial_t \rho(t, z_t) + \nabla_z^2 \rho(t, z_t) \, f(t, z_t)$$

$$= -\nabla_z [\nabla_z \cdot (\rho(t, z_t) \, f(t, z_t))] + \nabla_z^2 \rho(t, z_t) \, f(t, z_t) \tag{28}$$

$$= -\nabla_z [\nabla_z \rho(t, z_t)^\top f(t, z_t) + \rho(t, z_t) \, \nabla_z \cdot f(t, z_t)] + \nabla_z^2 \rho(t, z_t) \, f(t, z_t)$$

$$= -\nabla_z f(t, z_t)^\top \nabla_z \rho(t, z_t) - (\nabla_z \cdot f(t, z_t)) \nabla_z \rho(t, z_t) - \rho(t, z_t) \nabla_z (\nabla_z \cdot f(t, z_t)).$$

Therefore,

$$\partial_t s_t = \frac{\mathrm{d}}{\mathrm{d}t} [\nabla_z \log \rho(t, z_t)] = \frac{\mathrm{d}}{\mathrm{d}t} [\nabla_z \rho(t, z_t) / \rho(t, z_t)]$$

$$= \frac{\mathrm{d}}{\mathrm{d}t} (\nabla_z \rho(t, z_t)) / \rho(t, z_t) - \nabla_z \rho(t, z_t) \frac{\mathrm{d}}{\mathrm{d}t} (\rho(t, z_t)) / \rho(t, z_t)^2$$

$$= -\nabla_z f(t, z_t)^\top \nabla_z \log \rho(t, z_t) - \nabla_z \cdot f(t, z_t) \nabla_z \log \rho(t, z_t)$$

$$\quad - \nabla_z (\nabla_z \cdot f(t, z_t)) + \nabla_z \log \rho(t, z_t) \nabla_z \cdot f(t, z_t)$$

$$= -\nabla_z f(t, z_t)^\top \nabla_z \log \rho(t, z_t) - \nabla_z (\nabla_z \cdot f(t, z_t))$$

$$= -\nabla_z f(t, z_t)^\top s_t - \nabla_z (\nabla_z \cdot f(t, z_t)),$$

where we have used (28) and (4) in the third inequality.

Finally we prove (3d). In order to simplify notation, we will omit $(t, z_t)$ when there is no confusion and rewrite $\rho(t, z_t)$ and $f(t, z_t)$ as $\rho$ and $f$. We also denote $\partial_i$ as the partial derivative w.r.t. the $i$-th variable in $z$. We first compute $\frac{\mathrm{d}}{\mathrm{d}t} \nabla_z^2 \rho(t, z_t)$.

$$\frac{\mathrm{d}}{\mathrm{d}t} \nabla_z^2 \rho = \nabla_z^2 \partial_t \rho + \nabla_z^3 \rho \cdot f = -\nabla_z^2 [\nabla_z \cdot (\rho f)] + \nabla_z^3 \rho \cdot f$$

$$= -\nabla_z^2 [\nabla_z \rho^\top f + \rho \, \nabla_z \cdot f] + \nabla_z^3 \rho \cdot f = -\nabla_z^2 \left[ \sum_{i=1}^d \partial_i \rho \, f_i + \rho \, \nabla_z \cdot f \right] + \nabla_z^3 \rho \cdot f$$

$$= -\sum_{i=1}^d \left( \nabla_z^2 \partial_i \rho \, f_i + \partial_i \rho \, \nabla_z^2 f_i + \nabla_z \partial_i \rho \, \nabla_z f_i^\top + \nabla_z f_i \, \nabla_z \partial_i \rho^\top \right) \tag{29}$$

$$\quad - \left( \nabla_z^2 \rho \, (\nabla_z \cdot f) + \rho \, \nabla_z^2 (\nabla_z \cdot f) + \nabla_z \rho \, \nabla_z (\nabla_z \cdot f)^\top + \nabla_z (\nabla_z \cdot f) \, \nabla_z \rho^\top \right) + \nabla_z^3 \rho \cdot f$$

$$= -\sum_{i=1}^d \partial_i \rho \, \nabla_z^2 f_i - \nabla_z^2 \rho \, \nabla_z f - (\nabla_z f)^\top \nabla_z^2 \rho - \nabla_z^2 \rho \, (\nabla_z \cdot f) - \rho \nabla_z^2 (\nabla_z \cdot f)$$

$$\quad - \nabla_z \rho \, \nabla_z (\nabla_z \cdot f)^\top - \nabla_z (\nabla_z \cdot f) \nabla_z \rho^\top.$$

We also have

$$\nabla_z^2 \log(\rho) = \nabla_z \left( \frac{\nabla_z \rho}{\rho} \right) = \frac{\nabla_z^2 \rho}{\rho} - \frac{\nabla_z \rho \, \nabla_z \rho^\top}{\rho^2}, \tag{30}$$

which implies

$$\frac{\nabla_z^2 \rho}{\rho} = H_t + s_t s_t^\top \,. \tag{31}$$

Dividing (29) by $\rho$, we obtain

$$\frac{\frac{\mathrm{d}}{\mathrm{d}t} \nabla_z^2 \rho}{\rho}$$

$$= -\sum_{i=1}^d \frac{\partial_i \rho}{\rho} \, \nabla_z^2 f_i - \frac{\nabla_z^2 \rho}{\rho} \, \nabla_z f - (\nabla_z f)^\top \frac{\nabla_z^2 \rho}{\rho} - \frac{\nabla_z^2 \rho}{\rho} \, (\nabla_z \cdot f)$$

$$\quad - \nabla_z^2 (\nabla_z \cdot f) - \frac{\nabla_z \rho}{\rho} \, \nabla_z (\nabla_z \cdot f)^\top - \nabla_z (\nabla_z \cdot f) \frac{\nabla_z \rho^\top}{\rho} \tag{32}$$

$$= -\sum_{i=1}^d s_{it} \nabla_z^2 f_i - \left( H_t + s_t s_t^\top \right) \nabla_z f - (\nabla_z f)^\top \left( H_t + s_t s_t^\top \right)$$

$$\quad - (\nabla_z \cdot f) \left( H_t + s_t s_t^\top \right) - \nabla_z^2 (\nabla_z \cdot f) - s_t \nabla_z (\nabla_z \cdot f)^\top - \nabla_z (\nabla_z \cdot f) s_t^\top \,,$$

where we have used equation (31) in the second equality. Finally, we get

$$\partial_t H_t = \frac{\mathrm{d}}{\mathrm{d}t} \nabla_z^2 \log \rho = \frac{\mathrm{d}}{\mathrm{d}t} \left( \frac{\nabla_z^2 \rho}{\rho} - \frac{\nabla_z \rho \, \nabla_z \rho^\top}{\rho^2} \right)$$

$$= \frac{\frac{\mathrm{d}}{\mathrm{d}t} \nabla_z^2 \rho}{\rho} - \frac{\nabla_z^2 \rho \frac{\mathrm{d}}{\mathrm{d}t} \rho}{\rho^2} - \frac{\left( \frac{\mathrm{d}}{\mathrm{d}t} \nabla_z \rho \right) \nabla_z \rho^\top + \nabla_z \rho \left( \frac{\mathrm{d}}{\mathrm{d}t} \nabla_z \rho \right)^\top}{\rho^2} + \frac{2 \nabla_z \rho \, \nabla_z \rho^\top \frac{\mathrm{d}}{\mathrm{d}t} \rho}{\rho^3}$$

$$= -\sum_{i=1}^d s_{it} \nabla_z^2 f_i - \left( H_t + s_t s_t^\top \right) \nabla_z f - (\nabla_z f)^\top \left( H_t + s_t s_t^\top \right) - (\nabla_z \cdot f) \left( H_t + s_t s_t^\top \right)$$

$$\quad - \nabla_z^2 (\nabla_z \cdot f) - s_t \nabla_z (\nabla_z \cdot f)^\top - \nabla_z (\nabla_z \cdot f) s_t^\top + \left( H_t + s_t s_t^\top \right) (\nabla_z \cdot f)$$

$$\quad + \left( \nabla_z f^\top s_t + (\nabla_z \cdot f) s_t + \nabla_z (\nabla_z \cdot f) \right) s_t^\top$$

$$\quad + s_t \left( \nabla_z f^\top s_t + (\nabla_z \cdot f) s_t + \nabla_z (\nabla_z \cdot f) \right)^\top - 2 s_t s_t^\top (\nabla_z \cdot f)$$

$$= -\sum_{i=1}^d s_{it} \nabla_z^2 f_i - \nabla_z^2 (\nabla_z \cdot f) - H_t \nabla_z f - (\nabla_z f)^\top H_t \,.$$

We have used (30) in the second equality and used (32), (31), (4), and (28) in the fourth equality. This finishes the proof. □

**Proposition 2** (Information equality). *The following equality holds:*

$$\mathbb{E} \left[ \mathrm{Tr}(H_t) \right] = -\mathbb{E} \left[ |s_t|^2 \right] \,, \quad \textit{for all } t \geq 0 \,.$$

*Proof.* For all $t \geq 0$, we have

$$\mathbb{E} \left[ \mathrm{Tr}(H_t) + |s_t|^2 \right] = \mathbb{E} \left[ \nabla_z \cdot (\nabla_z \log \rho(t, z_t)) + |\nabla_z \log \rho(t, z_t)|^2 \right]$$

$$= \int_{\mathbb{R}^d} \left( \nabla_x \cdot (\nabla_x \log \rho(t, x)) + |\nabla_x \log \rho(t, x)|^2 \right) \rho(t, x) \, \mathrm{d}x$$

$$= \int_{\mathbb{R}^d} \left[ \nabla_x \cdot \left( \frac{\nabla_x \rho(t, x)}{\rho(t, x)} \right) \rho(t, x) + \frac{|\nabla_x \rho(t, x)|^2}{\rho(t, x)^2} \rho(t, x) \right] \mathrm{d}x$$

$$= \int_{\mathbb{R}^d} \left[ -\frac{\nabla_x \rho(t, x)^\top}{\rho(t, x)} \nabla_x \rho(t, x) + \frac{|\nabla_x \rho(t, x)|^2}{\rho(t, x)} \right] \mathrm{d}x = 0 \,,$$

where we have used the integration by part in the second last equality. □

**Covariance evolution of centered Gaussian distributions.** Let $z_0$ follow a centered Gaussian distribution $N(0, \Sigma(0))$ and $\partial_t z_t = A(t) z_t$. Then $z_t$ is also a centered Gaussian distribution $N(0, \Sigma(t))$, where the covariance $\Sigma(t)$ satisfies a matrix ODE:

$$\partial_t \Sigma(t) = A(t)\Sigma(t) + \Sigma(t) A(t)^\top . \tag{33}$$

*Proof.* We write down the density function $\rho(t, x)$ for $N(0, \Sigma(t))$

$$\rho(t, x) = (2\pi)^{-\frac{d}{2}} \det(\Sigma(t))^{-\frac{1}{2}} \exp\left(-\frac{1}{2} x^\top \Sigma(t)^{-1} x\right) .$$

It is sufficient to check that $\rho(t, x)$ satisfies the transport equation

$$\partial_t \rho(t, x) + \nabla_x \cdot (A(t)x\, \rho(t, x)) = 0 ,$$

if equation (33) holds. We first compute

$$\frac{\mathrm{d}}{\mathrm{d}t} \det(\Sigma(t)) = \det(\Sigma(t)) \operatorname{Tr}\left(\Sigma(t)^{-1} \partial_t \Sigma(t)\right) = \det(\Sigma(t)) \operatorname{Tr}\left(\Sigma(t)^{-1} \left(A(t)\Sigma(t) + \Sigma(t)A(t)^\top\right)\right)$$

$$= \det(\Sigma(t)) \operatorname{Tr}\left(\Sigma(t)^{-1} A(t)\Sigma(t) + A(t)^\top\right) = 2 \det(\Sigma(t)) \operatorname{Tr}(A(t)) ,$$

which implies

$$\frac{\mathrm{d}}{\mathrm{d}t}\left(\det(\Sigma(t))^{-\frac{1}{2}}\right) = -\frac{1}{2} \det(\Sigma(t))^{-\frac{3}{2}} \frac{\mathrm{d}}{\mathrm{d}t} \det(\Sigma(t)) = -\det(\Sigma(t))^{-\frac{1}{2}} \operatorname{Tr}(A(t)) . \tag{34}$$

We also have

$$\frac{\mathrm{d}}{\mathrm{d}t}\left(\Sigma(t)^{-1}\right) = -\Sigma(t)^{-1} \partial_t \Sigma(t)\, \Sigma(t)^{-1} = -\Sigma(t)^{-1} A(t) - A(t)^\top \Sigma(t)^{-1} ,$$

which implies

$$\frac{\mathrm{d}}{\mathrm{d}t} \exp\left(-\frac{1}{2} x^\top \Sigma(t)^{-1} x\right) = -\frac{1}{2} x^\top \frac{\mathrm{d}}{\mathrm{d}t}\left(\Sigma(t)^{-1}\right) x \exp\left(-\frac{1}{2} x^\top \Sigma(t)^{-1} x\right)$$

$$= \frac{1}{2} x^\top \left(\Sigma(t)^{-1} A(t) + A(t)^\top \Sigma(t)^{-1}\right) x \exp\left(-\frac{1}{2} x^\top \Sigma(t)^{-1} x\right) \tag{35}$$

$$= x^\top A(t)^\top \Sigma(t)^{-1} x \exp\left(-\frac{1}{2} x^\top \Sigma(t)^{-1} x\right) .$$

Combining (34) and (35), we obtain

$$\partial_t \rho(t, x) = (2\pi)^{-\frac{d}{2}} \frac{\mathrm{d}}{\mathrm{d}t}\left(\det(\Sigma(t))^{-\frac{1}{2}}\right) \exp\left(-\frac{1}{2} x^\top \Sigma(t)^{-1} x\right)$$

$$+ (2\pi)^{-\frac{d}{2}} \det(\Sigma(t))^{-\frac{1}{2}} \frac{\mathrm{d}}{\mathrm{d}t} \exp\left(-\frac{1}{2} x^\top \Sigma(t)^{-1} x\right)$$

$$= \rho(t, x)\left[-\operatorname{Tr}(A(t)) + x^\top A(t)^\top \Sigma(t)^{-1} x\right] .$$

Finally, we obtain

$$\nabla_x \cdot (A(t)x\, \rho(t, x)) = \operatorname{Tr}(A(t))\, \rho(t, x) + x^\top A(t)^\top \nabla_x \rho(t, x)$$

$$= \operatorname{Tr}(A(t))\, \rho(t, x) + x^\top A(t)^\top \left(-\Sigma(t)^{-1} x\right) \rho(t, x) = -\partial_t \rho(t, x) ,$$

which finishes the proof. $\qquad\square$

## A.2 Details for the MFC problems

We recall the formulation of the MFC problem

$$\inf_v \int_0^{t_{\text{end}}} \int_{\mathbb{R}^d} [L(t, x, v(t, x)) + F(t, x, \rho(t, x))]\, \rho(t, x)\, \mathrm{d}x\, \mathrm{d}t + \int_{\mathbb{R}^d} G(x, \rho(t_{\text{end}}, x)) \rho(t_{\text{end}}, x)\, \mathrm{d}x ,$$

where the density $\rho(t,x)$ satisfies the FP equation with a given initialization

$$\partial_t \rho(t,x) + \nabla_x \cdot (\rho(t,x)v(t,x)) = \gamma\Delta_x\rho(t,x)\,, \quad \rho(0,x) = \rho_0(x)\,.$$

This FP equation has a stochastic characterization

$$\mathrm{d}x_t = v(t,x_t)\,\mathrm{d}t + \sqrt{2\gamma}\,\mathrm{d}W_t\,,$$

where $W_t$ is a standard Brownian motion. For the well-posedness of the problem, we assume that the running cost $L$ is strongly convex in $v$, then the Hamiltonian, i.e. the Legendre transform (convex conjugate) of $L$, is well defined, given by

$$H(t,x,p) = \sup_{v\in\mathbb{R}^d} v^\top p - L(t,x,v)\,.$$

According to standard results in convex analysis, $\mathrm{D}_v L$ and $\mathrm{D}_p H$ are the inverse functions to each other, in the sense that

$$\begin{aligned} \mathrm{D}_p H(t,x,\mathrm{D}_v L(t,x,v)) &= v\,, \quad \forall v\,, \\ \mathrm{D}_v L(t,x,\mathrm{D}_p H(t,x,p)) &= p\,, \quad \forall p\,. \end{aligned} \tag{36}$$

We recall that the composed velocity is

$$f(t,x) = v(t,x) - \gamma\nabla_x\log\rho(t,x)\,,$$

and the probability flow $z_t$ is given by

$$\partial_t z_t = f(t,z_t)\,.$$

Note that the stochastic dynamic $x_t$ and the probability flow $z_t$ share the same probability distribution, while they are different dynamics. This $z_t$ serves as a characteristic line for the system. Along this line, a forward Euler scheme gives $\mathcal{O}(\Delta t)$ error, while a direct discretization of the stochastic dynamic has an error term $\mathcal{O}(\sqrt{\Delta t})$.

Next, we present the an analysis for the modified HJB equation of MFC problem, tailored for the probability flow dynamic.

**Proposition 3.** *Let $L$ be strongly convex in $v$, then solution to the MFC problem* (17) *is as follows. Consider a function $\psi\colon [0,t_{end}]\times\mathbb{R}^d \to \mathbb{R}$, such that*

$$f(t,x) = \mathrm{D}_p H(t,x,\nabla_x\psi(t,x) + \gamma\nabla_x\log\rho(t,x)) - \gamma\nabla_x\log\rho(t,x),$$

*where the density function $\rho(t,x)$ and $\psi\colon [0,t_{end}]\times\mathbb{R}^d \to \mathbb{R}$ satisfy the following system of equations*

$$\begin{cases} \partial_t\rho(t,x) + \nabla_x \cdot (\rho(t,x)\mathrm{D}_p H) = \gamma\Delta_x\rho(t,x), \\ \partial_t\psi(t,x) + \nabla_x\psi(t,x)^\top \mathrm{D}_p H - \gamma\nabla_x\cdot \mathrm{D}_p H + \gamma\Delta_x\psi(t,x) - L(t,x,\mathrm{D}_p H) \\ \quad - \widetilde{F}(t,x,\rho(t,x)) + 2\gamma^2\Delta_x\log\rho(t,x) + \gamma^2\left|\nabla_x\log\rho(t,x)\right|^2 = 0, \\ \rho(0,x) = \rho_0(x), \quad \psi(t_{end},x) = -\widetilde{G}(x,\rho(t_{end},x)) - \gamma\log\rho(t_{end},x), \end{cases}$$

*Here, $\mathrm{D}_p H$ is short for $\mathrm{D}_p H(t,x,\nabla_x\psi(t,x) + \gamma\nabla_x\log\rho(t,x))$, $\widetilde{F}(t,x,\rho) = \dfrac{\partial}{\partial\rho}(F(t,x,\rho)\rho) = \dfrac{\partial F}{\partial\rho}(t,x,\rho)\rho + F(t,x,\rho)$, and $\widetilde{G}(x,\rho) = \dfrac{\partial G}{\partial\rho}(x,\rho)\rho + G(x,\rho)$.*

*Proof.* We start by adding a Lagrange multiplier $\phi(t,x)$ and obtain the augmented objective

$$\int_0^{t_{end}}\int_{\mathbb{R}^d} \left[\left(L(t,x,f(t,x) + \gamma\nabla_x\log\rho(t,x)) + F(t,x,\rho(t,x))\right)\rho(t,x) \right.$$

$$\left. + \phi(t,x)\left(\partial_t\rho(t,x) + \nabla_x\cdot(f(t,x)\rho(t,x))\right)\right]\mathrm{d}x\,\mathrm{d}t + \int_{\mathbb{R}^d} G(x,\rho(t_{end},x))\rho(t_{end},x)\,\mathrm{d}x$$

$$= \int_0^{t_{end}}\int_{\mathbb{R}^d}\left[\left(L(t,x,f(t,x) + \gamma\nabla_x\log\rho(t,x)) + F(t,x,\rho(t,x))\right)\rho(t,x)\right.$$

$$\left. - \partial_t\phi(t,x)\rho(t,x) - \nabla_x\phi(t,x)^\top f(t,x)\rho(t,x)\right]\mathrm{d}x\,\mathrm{d}t \tag{37}$$

$$+ \int_{\mathbb{R}^d}(G(x,\rho(t_{end},x)) + \phi(t_{end},x))\rho(t_{end},x)\,\mathrm{d}x - \int_{\mathbb{R}^d}\phi(0,x)\rho(0,x)\,\mathrm{d}x.$$

Taking the variation of (37) w.r.t. $\rho(t_{\text{end}}, \cdot)$, we get $\phi(t_{\text{end}}, x) = -\widetilde{G}(x, \rho(t_{\text{end}}, x))$. Taking the variation w.r.t. $f$, we obtain

$$D_v L(t, x, f(t, x) + \gamma \nabla_x \log \rho(t, x)) = \nabla_x \phi(t, x) \,,$$

which implies

$$f(t, x) + \gamma \nabla_x \log \rho(t, x) = D_p H(t, x, \nabla_x \phi(t, x)) \,, \tag{38}$$

due to (36). Next, we denote $D_p H(t, x, \nabla_x \phi(t, x))$ by $D_p H$ for short. We then observe that

$$D_v L := D_v L(t, x, f(t, x) + \gamma \nabla_x \log \rho(t, x)) = D_v L(t, x, D_p H) = \nabla_x \phi(t, x).$$

Taking the variation of (37) w.r.t. $\rho(\cdot, \cdot)$, we get

$$\begin{aligned}
0 &= -\gamma \nabla_x \cdot [D_v L \, \rho(t, x)] / \rho(t, x) + L(t, x, f(t, x) + \gamma \nabla_x \log \rho(t, x)) \\
&\quad + \widetilde{F}(t, x, \rho(t, x)) - \partial_t \phi(t, x) - \nabla_x \phi(t, x)^\top f(t, x) \\
&= -\gamma \nabla_x \cdot D_v L - \gamma D_v L^\top \nabla_x \log \rho(t, x) + L(t, x, D_p H) \\
&\quad + \widetilde{F}(t, x, \rho(t, x)) - \partial_t \phi(t, x) - \nabla_x \phi(t, x)^\top (D_p H - \gamma \nabla_x \log \rho(t, x)) \\
&= -\gamma \Delta_x \phi(t, x) + L(t, x, D_p H) + \widetilde{F}(t, x, \rho(t, x)) - \partial_t \phi(t, x) - \nabla_x \phi(t, x)^\top D_p H \,, \tag{39}
\end{aligned}$$

where we have used $D_v L = \nabla_x \phi(t, x)$ in the last inequality. Equation (39) is the classical HJB equation corresponding to the MFC problem, which coincide with equation (4.6) in Bensoussan et al. (2013) after a sign flip, where our $D_p H$ is their $\hat{v}$. Please note that, throughout our derivation, $\rho(t, x)$ is not a general density, but the density under the optimal control.

We next present the modified HJB equation. We define

$$\psi(t, x) := \phi(t, x) - \gamma \log \rho(t, x) \,. \tag{40}$$

This definition seems to complicate the system, but eventually gives us an elegant characterization, especially in the case of LQ problem (see Corollary 1). We then observe that

$$\begin{aligned}
\partial_t \log \rho(t, x) &= \frac{\partial_t \rho(t, x)}{\rho(t, x)} = -\frac{\nabla_x \cdot (\rho(t, x) f(t, x))}{\rho(t, x)} \\
&= -\nabla_x \cdot f(t, x) - \nabla_x \log \rho(t, x)^\top f(t, x) \\
&= -\nabla_x \cdot D_p H + \gamma \Delta_x \log \rho(t, x) - \nabla_x \log \rho(t, x)^\top D_p H + \gamma \left| \nabla_x \log \rho(t, x) \right|^2 \,, \tag{41}
\end{aligned}$$

where the last equality is due to (38). Therefore, the HJB equation (39) is transformed into

$$\begin{aligned}
0 &= -\gamma \Delta_x \psi(t, x) - \gamma^2 \Delta_x \log \rho(t, x) + L(t, x, D_p H) + \widetilde{F}(t, x, \rho(t, x)) - \partial_t \psi(t, x) \\
&\quad - \gamma \partial_t \log \rho(t, x) - (\nabla_x \psi(t, x) + \gamma \nabla_x \log \rho(t, x))^\top D_p H \\
&= -\gamma \Delta_x \psi(t, x) - \gamma^2 \Delta_x \log \rho(t, x) + L(t, x, D_p H) + \widetilde{F}(t, x, \rho(t, x)) - \partial_t \psi(t, x) \\
&\quad + \gamma \nabla_x \cdot D_p H - \gamma^2 \Delta_x \log \rho(t, x) + \gamma \nabla_x \log \rho(t, x)^\top D_p H - \gamma^2 \left| \nabla_x \log \rho(t, x) \right|^2 \\
&\quad - (\nabla_x \psi(t, x) + \gamma \nabla_x \log \rho(t, x))^\top D_p H \\
&= -\partial_t \psi(t, x) - \nabla_x \psi(t, x)^\top D_p H + \gamma \nabla_x \cdot D_p H - \gamma \Delta_x \psi(t, x) + L(t, x, D_p H) \\
&\quad + \widetilde{F}(t, x, \rho(t, x)) - 2\gamma^2 \Delta_x \log \rho(t, x) - \gamma^2 \left| \nabla_x \log \rho(t, x) \right|^2 \,.
\end{aligned}$$

Here, $D_p H$ is short for $D_p H(t, x, \nabla_x \psi(t, x) + \gamma \nabla_x \log \rho(t, x))$. We plug in the shift (40) in the first equality; we plug in (41) in the second equality. Therefore, we have recovered Proposition 3. $\quad\square$

**Corollary 1.** *In the LQ problem where $L(t, x, v) = \frac{1}{2}|v|^2$ and $F(t, x, \rho) = 0$, the solution of the MFC problem* (17) *is characterized by*

$$f(t, x) = \nabla_x \psi(t, x) \,,$$

*and*

$$\begin{cases}
\partial_t \rho(t, x) + \nabla_x \cdot (\rho(t, x) \nabla_x \psi(t, x)) = 0 \,, \\
\partial_t \psi(t, x) + \dfrac{1}{2} \left| \nabla_x \psi(t, x) \right|^2 + \gamma^2 \Delta_x \log \rho(t, x) + \dfrac{1}{2}\gamma^2 \left| \nabla_x \log \rho(t, x) \right|^2 = 0 \,, \\
\rho(0, x) = \rho_0(x) \,, \quad \psi(t_{end}, x) = -\widetilde{G}(x, \rho(t_{end}, x)) - \gamma \log \rho(t_{end}, x) \,.
\end{cases}$$

*If we further define the Fisher information as $I[\rho] := \int_{\mathbb{R}^d} |\nabla_x \log \rho(x)|^2 \rho(x)\,\mathrm{d}x$, then the equation for $\psi$ becomes*

$$\partial_t \psi(t,x) + \frac{1}{2}|\nabla_x \psi(t,x)|^2 - \frac{1}{2}\gamma^2 \frac{\delta I[\rho(t,\cdot)]}{\delta \rho(t,\cdot)}(x) = 0\,.$$

*Proof.* When $L(t,x,v) = \frac{1}{2}|v|^2$, we have $\mathrm{D}_v L(t,x,v) = v$ and $\mathrm{D}_p H(t,x,p) = p$. So

$$\mathrm{D}_p H(t,x,\nabla_x \psi(t,x) + \gamma \nabla_x \log \rho(t,x)) = \nabla_x \psi(t,x) + \gamma \nabla_x \log \rho(t,x)\,.$$

Hence, the equation for $\psi$ becomes

$$0 = \partial_t \psi(t,x) + \nabla_x \psi(t,x)^\top \left(\nabla_x \psi(t,x) + \gamma \nabla_x \log \rho(t,x)\right) - \gamma \Delta_x \psi(t,x)$$

$$- \gamma^2 \Delta_x \log \rho(t,x) + \gamma \Delta_x \psi(t,x) - \frac{1}{2}|\nabla_x \psi(t,x) + \gamma \nabla_x \log \rho(t,x)|^2$$

$$+ 2\gamma^2 \Delta_x \log \rho(t,x) + \gamma^2 |\nabla_x \log \rho(t,x)|^2$$

$$= \partial_t \psi(t,x) + \frac{1}{2}|\nabla_x \psi(t,x)|^2 + \gamma^2 \Delta_x \log \rho(t,x) + \frac{1}{2}\gamma^2 |\nabla_x \log \rho(t,x)|^2\,.$$

A direct computation gives

$$\frac{\delta I[\rho]}{\delta \rho}(x) = -2\Delta_x \log \rho(x) - |\nabla_x \log \rho(x)|^2\,,$$

which implies

$$\partial_t \psi(t,x) + \frac{1}{2}|\nabla_x \psi(t,x)|^2 - \frac{1}{2}\gamma^2 \frac{\delta I[\rho(t,\cdot)]}{\delta \rho(t,\cdot)}(x) = 0\,.$$

$\square$

### A.3 Formulation for flow matching of overdamped Langevin dynamic

In this section, we present the flow matching problem of overdamped Langevin dynamic

$$\mathrm{d}X_t = b(t,X_t)\,\mathrm{d}t + \sqrt{2\gamma}\,\mathrm{d}W_t\,, \qquad X_0 \sim \rho_0\,.$$

Here we assume that the drift is the negative gradient of some potential function

$$b(t,x) = -\nabla_x V(x)\,,$$

so that the stationary distribution for the overdamped Langevin dynamic is proportional to $\exp(-V(x)/\gamma)$. The goal for flow matching is to learn a velocity field $v(x)$ that matches $b(t,x)$. One minimizes the objective functional

$$\inf_v \int_0^{t_{\text{end}}} \int_{\mathbb{R}^d} \frac{1}{2}|v(t,x) - b(t,x)|^2 \rho(t,x)\,\mathrm{d}x\,\mathrm{d}t\,, \tag{42}$$

subject to the FP equation

$$\partial_t \rho(t,x) + \nabla_x \cdot (\rho(t,x)v(t,x)) = \gamma \Delta_x \rho(t,x)\,, \quad \rho(0,x) = \rho_0(x)\,. \tag{43}$$

The following corollary gives the exact formulation for the solution.

**Corollary 2.** *For flow matching problem of overdamped Langevin dynamic where $L(t,x,v) = \frac{1}{2}|v - b(t,x)|^2$, $F(t,x,\rho) = 0$ and $G(x,\rho) = 0$, the solution of the MFC problem* (17) *is characterized by*

$$f(t,x) = \nabla_x \psi(t,x) + b(t,x)\,,$$

*and*

$$\begin{cases} \partial_t \rho(t,x) + \nabla_x \cdot (\rho(t,x)(\nabla_x \psi(t,x) + b(t,x))) = 0\,, \\[2mm] \partial_t \psi(t,x) + \frac{1}{2}|\nabla_x \psi(t,x)|^2 + \nabla_x \psi(t,x)^\top b(t,x) - \gamma \nabla_x \cdot b(t,x) \\[2mm] \qquad + \gamma^2 \Delta_x \log \rho(t,x) + \frac{1}{2}\gamma^2 |\nabla_x \log \rho(t,x)|^2 = 0\,, \\[2mm] \rho(0,x) = \rho_0(x)\,, \quad \psi(t_{\text{end}},x) = -\gamma \log \rho(t_{\text{end}},x)\,. \end{cases}$$

*Proof.* Through direct computation, the convex conjugate of $L$ is

$$H(t,x,p) = \frac{1}{2}|p|^2 + p^\top b(t,x)\,,$$

and $\mathrm{D}_p H(t,x,p) = p + b(t,x)$. Plugging these expressions into Proposition 3, we recover the results for this corollary. $\square$

# B DETAILS FOR NUMERICAL IMPLEMENTATION

## B.1 HJB REGULARIZER FOR MFC

In addition to the numerical algorithm in section 4.3, we can add the residual of the HJB equation (20) into the loss function. First, let us consider the HJB regularizer in the LQ Gaussian case, as presented in Corollary 1. Here the Gaussian case means that $\rho_0$ is also given as a Gaussian distribution, in which the solution of density function in LQ MFC problem stays in a time dependent Gaussian distribution.

In the LQ scenario, parametrizing the vector field $f$ directly as a neural network complicates the system, as terms like $\partial_t \psi$ in (22) are hard to implement. A more feasible approach is to parametrize $\psi$ as a neural network, allowing the vector field $f$ to be computed as $f(t, x) = \nabla_x \psi(t, x)$.

This parametrization, however, requires two additional orders of auto-differentiations compared to the original algorithm. First, if $\psi$ is parametrized as a neural network as in (5), then $f(t, x) = \nabla_x \psi(t, x)$ must be obtained through auto-differentiation. Second, the term $\Delta_x \rho(t, x)$ in (22) requires computing the dynamic for $H_t$ through (10), which requires another order of derivative. This term $H_t$ is not required in the original Algorithm 1. These two additional differentiations could potentially complicate the optimization landscape.

Therefore, in order to make a clean and fair comparison in studying the effect of the HJB regularizer, we consider the specific parametrization for $\psi$, given by

$$\psi(t, x) = \frac{1}{2} x^\top A(t) x + B(t)^\top x + C(t), \tag{44}$$

where $A(t) : [0, t_{\text{end}}] \to \mathbb{R}^{d \times d}$, $B(t) : [0, t_{\text{end}}] \to \mathbb{R}^d$, and $C(t) : [0, t_{\text{end}}] \to \mathbb{R}$. These three functions can be further parametrized into trainable variables $\theta = [\theta^A, \theta^B, \theta^C]$ with $\theta^A \in \mathbb{R}_{\text{sym}}^{(N_t+1) \times d \times d}$, $\theta^B \in \mathbb{R}^{(N_t+1) \times d}$, and $\theta^C \in \mathbb{R}^{N_t+1}$ after time discretizations. I.e., we only evaluate $\psi$ when $t = t_j$ ($j = 0, 1, \ldots, N_t$). Here, $\theta^A \in \mathbb{R}_{\text{sym}}^{(N_t+1) \times d \times d}$ means that each slice of matrix $\theta_j^A \in \mathbb{R}^{d \times d}$ is symmetric. This parametrization is different from (5), but the numerical implementation for the ODE discretization (6)-(10) and loss cost (24) remain unchanged, with the vector field defined by

$$f(t_j, x; \theta) = \nabla_x \psi(t, x; \theta) = \theta_j^A x + \theta_j^B. \tag{45}$$

In the LQ example with Gaussian distribution, the solution is indeed of this form; see Bensoussan et al. (2013) Chapter 6 for details.

With such parametrizations, we can compute the residual of the HJB equation through (23) as a regularizer. Here, the term $\partial_t \psi(t, z_t)$ in (23) could be approximated using the finite difference scheme. In this work, we use the central difference scheme

$$\begin{aligned}
\partial_t \psi(t_j, x; \theta) &\approx \frac{1}{2\Delta t} \left( \psi(t_{j+1}, x; \theta) - \psi(t_{j-1}, x; \theta) \right) \\
&= \frac{1}{2\Delta t} \left( \frac{1}{2} x^\top (\theta_{j+1}^A - \theta_{j-1}^A) x + (\theta_{j+1}^B - \theta_{j-1}^B)^\top x + (\theta_{j+1}^C - \theta_{j-1}^C) \right).
\end{aligned} \tag{46}$$

With this discretization, the numerical residual of the HJB equation at $t_j$ (cf. (23)) is given by

$$\begin{aligned}
\mathcal{L}_{\text{HJB}t_j} = \frac{1}{N_z} \sum_{n=1}^{N_z} \Bigg[ &\frac{1}{2\Delta t} \left( \psi(t_{j+1}, z_{t_j}^{(n)}; \theta) - \psi(t_{j-1}, z_{t_j}^{(n)}; \theta) \right) + \frac{1}{2} \left| \nabla_z \psi(t_j, z_{t_j}^{(n)}; \theta) \right|^2 \\
&+ \gamma^2 \left( \text{Tr}\left( H_{t_j}^{(n)} \right) + \frac{1}{2} \left| s_{t_j}^{(n)} \right|^2 \right) \Bigg],
\end{aligned} \tag{47}$$

and the total regularization loss is

$$\mathcal{L}_{\text{HJB}} = \sum_{j=1}^{N_t-1} \left| \mathcal{L}_{\text{HJB}t_j} \right| \Delta t. \tag{48}$$

Finally, we can minimize $\mathcal{L}_{\text{total}} = \mathcal{L}_{\text{cost}} + \lambda \mathcal{L}_{\text{HJB}}$ w.r.t. $\theta$ to solve the MFC problem, where $\lambda > 0$ is a weight parameter. We summarize the regularized method in Appendix B.2 Algorithm 2

**Regularized algorithm for flow matching of overdamped Langevin dynamics** This regularized algorithm could be extended to a more general setting, such as the flow matching problem for an OU process. We recall that the formulation for this problem is presented in Appendix A.3.

We consider the flow matching of OU process with a linear drift function $b(t, x) = -ax$ and an initial Gaussian distribution $N(\mu(0), \Sigma(0))$ where $\mu(0) \in \mathbb{R}^d$ and $\Sigma(0) \in \mathbb{R}^{d \times d}_{\text{sym}}$. Then the stochastic state dynamic is given by

$$\mathrm{d}x_t = -ax_t\,\mathrm{d}t + \sqrt{2\gamma}\,\mathrm{d}W_t\,, \qquad x_0 \sim N(\mu(0), \Sigma(0))\,.$$

Standard calculations inform us that $x_t \sim (\mu(t), \Sigma(t))$ remains to be Gaussian, where the mean and variance evolve as

$$\mu(t) = \exp(-at)\,\mu(0)\,,$$

and

$$\Sigma(t) = \frac{\gamma}{a}I_d + \left(\Sigma(0) - \frac{\gamma}{a}I_d\right)\exp(-2at)\,.$$

This expression provides a reference solution in the numerical test. In this work, we set $a = 1$, $\mu(0)$ to be an all-one vector in $\mathbb{R}^d$, and $\Sigma(0) = 4I_d$.

Similar to the algorithm presented above, we parametrize $\psi$ through (44). Please note that the solution $\psi$ to the MFC problem is given by Corollary 2. So, the vector field is

$$f(t_j, x; \theta) = \nabla_x \psi(t_j, x; \theta) + b(t_j, x) = \theta_j^A x + \theta_j^B - ax\,. \tag{49}$$

Also, the HJB equation along the state trajectory is written as

$$\partial_t \psi(t, z_t) + \frac{1}{2}\left|\nabla_z \psi(t, z_t)\right|^2 + \nabla_z \psi(t, z_t)^\top b(t, z_t) - \gamma \nabla_z \cdot b(t, t) + \gamma^2 \left(\mathrm{Tr}(H_t) + \frac{1}{2}|s_t|^2\right) = 0\,. \tag{50}$$

One can view (50) as an analog of (23) in the flow matching example.

The time discretization for the HJB equation is the same as (46), and the residual of the HJB equation at $t_j$ is given by

$$\mathcal{L}_{\mathrm{HJB}t_j} = \frac{1}{N_z}\sum_{n=1}^{N_z}\left[\frac{1}{2\Delta t}\left(\psi(t_{j+1}, z_{t_j}^{(n)}; \theta) - \psi(t_{j-1}, z_{t_j}^{(n)}; \theta)\right) + \frac{1}{2}\left|\nabla_z \psi(t_j, z_{t_j}^{(n)}; \theta)\right|^2\right.$$
$$\left. + \nabla_z \psi(t_j, z_{t_j}^{(n)}; \theta)^\top b(t_j, z_{t_j}^{(n)}) - \gamma \nabla_z \cdot b(t_j, z_{t_j}^{(n)}) + \gamma^2\left(\mathrm{Tr}\left(H_{t_j}^{(n)}\right) + \frac{1}{2}\left|s_{t_j}^{(n)}\right|^2\right)\right]\,, \tag{51}$$

and the total regularization loss is still

$$\mathcal{L}_{\mathrm{HJB}} = \sum_{j=1}^{N_t-1}\left|\mathcal{L}_{\mathrm{HJB}t_j}\right|\Delta t\,.$$

And the total loss is again $\mathcal{L}_{\text{total}} = \mathcal{L}_{\text{cost}} + \lambda \mathcal{L}_{\mathrm{HJB}}$. We summarize this method in Algorithm 3 Appendix B.2.

### B.2 PSEUDO-CODE FOR ALGORITHMS

We present all pseudo-codes for all algorithms in this section. Recall that the standard version of the MFC solver Algorithm 1 is presented in the main text. We present the regularized MFC solver for LQ problem Algorithm 2; the regularized MFC solver for the flow matching of OU process Algorithm 3; and the multi-stage splicing method Algorithm 4.

---

**Algorithm 2** Regularized score-based normalizing flow solver for the MFC problem

---

**Input:** MFC problem (21) (18), $N_t$, $N_z$, learning rate, weight parameter $\lambda$, number of iterations
**Output:** the solution to the MFC problem
    Initialize $\theta$
    **for** index $= 1$ **to** index$_{\text{end}}$ **do**
        Sample $N_z$ points $\{z_0^{(n)}\}_{n=1}^{N_z}$ from the initial distribution $\rho_0$
        Compute $\tilde{l}_0^{(n)} = \rho(0, z_0^{(n)})$, $s_0^{(n)} = \nabla_z \log \rho(0, z_0^{(n)})$, $H_0^{(n)} = \nabla_z^2 \log \rho(0, z_0^{(n)})$
        Initialize losses $\mathcal{L}_{\text{cost}} = 0$ and $\mathcal{L}_{\text{HJB}} = 0$
        **for** $j = 0$ **to** $N_t - 1$ **do**

$$\text{update loss } \mathcal{L}_{\text{cost}} \mathrel{+}= \frac{1}{N_z} \sum_{n=1}^{N_z} \frac{1}{2} \left| f(t_j, z_{t_j}^{(n)}; \theta) + \gamma s_{t_j}^{(n)} \right|^2 \Delta t$$

            compute $(\nabla_z, \nabla_z \cdot) f(t_j, z_{t_j}^{(n)}; \theta)$ through auto-differentiation
            **if** $j >= 1$ **then**
                Compute $\mathcal{L}_{\text{HJB}t_j}$ through (47)
                $\mathcal{L}_{\text{HJB}} \mathrel{+}= \left| \mathcal{L}_{\text{HJB}t_j} \right|$
            **end if**
            compute $z_{t_{j+1}}^{(n)}, \tilde{l}_{t_{j+1}}^{(n)}, s_{t_{j+1}}^{(n)}, H_{t_{j+1}}^{(n)}$ through the forward Euler scheme (6), (8), (9), and (10)
        **end for**

        add terminal cost $\mathcal{L}_{\text{cost}} \mathrel{+}= \dfrac{1}{N_z} \sum_{n=1}^{N_z} G(z_{t_{N_t}}^{(n)}, \tilde{l}_{t_{N_t}}^{(n)})$

        Total loss $\mathcal{L}_{\text{total}} = \mathcal{L}_{\text{cost}} + \lambda \mathcal{L}_{\text{HJB}}$
        update the parameters $\theta$ through Adam method to minimize the loss $\mathcal{L}_{\text{total}}$
    **end for**

---

**Algorithm 3** Regularized score-based normalizing flow solver for flow matching of OU process

---

**Input:** MFC problem (42) (43), $N_t$, $N_z$, learning rate, weight parameter $\lambda$, number of iterations
**Output:** the solution to the MFC problem
    Initialize $\theta$
    **for** index $= 1$ **to** index$_{\text{end}}$ **do**
        Sample $N_z$ points $\{z_0^{(n)}\}_{n=1}^{N_z}$ from the initial distribution $\rho_0$
        Compute $s_0^{(n)} = \nabla_z \log \rho(0, z_0^{(n)})$, $H_0^{(n)} = \nabla_z^2 \log \rho(0, z_0^{(n)})$
        Initialize losses $\mathcal{L}_{\text{cost}} = 0$ and $\mathcal{L}_{\text{HJB}} = 0$
        **for** $j = 0$ **to** $N_t - 1$ **do**

$$\text{update loss } \mathcal{L}_{\text{cost}} \mathrel{+}= \frac{1}{N_z} \sum_{n=1}^{N_z} \frac{1}{2} \left| f(t_j, z_{t_j}^{(n)}; \theta) + \gamma s_{t_j}^{(n)} - b(t_j, z_{t_j}^{(n)}) \right|^2 \Delta t$$

            compute $(\nabla_z, \nabla_z \cdot) f(t_j, z_{t_j}^{(n)}; \theta)$ through auto-differentiation
            **if** $j >= 1$ **then**
                Compute $\mathcal{L}_{\text{HJB}t_j}$ through (51)
                $\mathcal{L}_{\text{HJB}} \mathrel{+}= \left| \mathcal{L}_{\text{HJB}t_j} \right|$
            **end if**
            compute $z_{t_{j+1}}^{(n)}\ s_{t_{j+1}}^{(n)}, H_{t_{j+1}}^{(n)}$ through the forward Euler scheme (6), (9), and (10)
        **end for**
        Total loss $\mathcal{L}_{\text{total}} = \mathcal{L}_{\text{cost}} + \lambda \mathcal{L}_{\text{HJB}}$
        update the parameters $\theta$ through Adam method to minimize the loss $\mathcal{L}_{\text{total}}$
    **end for**

---

---

**Algorithm 4** Multi-stage splicing algorithm for MFC problems

---

**Input:** MFC problem (14) (15), $N_t$, $N_z$, network structure (5), learning rate, number of iterations, partition of the total time interval $\{I_m\}_{m=1}^{N_{\text{stage}}}$

**Output:** the solution to the MFC problem

   Initialize $\theta_0$ for the neural network

   Apply Algorithm 1 within the first interval $I_1$                         ▷ first stage training

   Save parameter of the network $\theta_1$ and the terminal states $\mathcal{S}_1 = \{z_{t_{N_t}}^{(n)}\}_{n=1}^{N_z}$ for the next stage

   **for** $m = 2$ **to** $N_{\text{stage}}$ **do**                                  ▷ follow-up stages training

      Load parameter $\theta_{m-1}$ for the network from the previous stage

      Apply Algorithm 1 within the interval $I_m$, but with a fixed set of initialization points $\mathcal{S}_{m-1}$ from the previous stage

      Save the parameter $\theta_m$ for the network and the terminal states $\mathcal{S}_m = \{z_{t_{N_t}}^{(n)}\}_{n=1}^{N_z}$ for the next stage

   **end for**

---

### B.3 THE ERRORS IN THE NUMERICAL EXAMPLES

**The errors for the standard Algorithm 1.** For examples with exact solutions, we present the errors for the density, the velocity field, and the score function. These numerical expressions are given by

$$\text{err}_\rho = \frac{1}{N_z} \sum_{n=1}^{N_z} \left| \tilde{l}_{t_{N_t}}^{(n)} - \rho(t_{N_t}, z_{t_{N_t}}^{(n)}) \right| \,,$$

$$\text{err}_f = \frac{1}{N_z} \frac{1}{N_t + 1} \sum_{n=1}^{N_z} \sum_{j=0}^{N_t} \left| f\left(t_j, z_{t_j}^{(n)}; \theta\right) - f\left(t_j, z_{t_j}^{(n)}\right) \right| \,, \tag{52}$$

$$\text{err}_s = \frac{1}{N_z} \sum_{n=1}^{N_z} \left| s_{t_{N_t}}^{(n)} - \nabla_z \log \rho\left(t_{N_t}, z_{t_{N_t}}^{(n)}\right) \right| \,.$$

The errors we present in the tables in section 5 and Appendix C are the average errors over 10 independent runs. Note that there are two sources that contribute to these three errors $\text{err}_\rho$, $\text{err}_f$, and $\text{err}_s$. We take the terminal time $t_{t_N} = t_{\text{end}}$ as an example. The first source is the discretization error for the state dynamic, characterized by $z_{t_{\text{end}}} - z_{t_{N_t}}$, where $z_{t_{\text{end}}}$ is the end point of the continuous dynamic (3a) and $z_{t_{N_t}}$ is its time discretization. The second source comes from the training error of the neural network $f(\cdot, \cdot; \theta)$. When a function is steep, especially the score function $\nabla \log \rho$ in a non-Gaussian example, a small discretization error could be magnified by a large factor due to this large condition number. This is the reason why the score errors for the double well potential function example in Appendix C.4 are larger compared with other results.

**The errors for the double well potential example.** For the double well potential example, it is hard to obtain a reference solution for $t \in (0, t_{\text{end}})$ (cf. Appendix C.4). As an alternative, we compute the error of the velocity field at initial and terminal time $t = 0, t_{\text{end}}$ through

$$\text{err}_{f_0} = \frac{1}{N_z} \sum_{n=1}^{N_z} \left| f\left(0, z_0^{(n)}; \theta\right) - f\left(0, z_0^{(n)}\right) \right| \,,$$

$$\tag{53}$$

$$\text{err}_{f_{end}} = \frac{1}{N_z} \sum_{n=1}^{N_z} \left| f\left(t_{N_t}, z_{t_{N_t}}^{(n)}; \theta\right) - f\left(t_{N_t}, z_{t_{N_t}}^{(n)}\right) \right| \,.$$

**The errors for the regularized Algorithm 2 and 3.** For modified algorithms with HJB regularizers, including Algorithm 2 and 3, we compute the error of the parameter $A$ and $B$ directly, given by

$$\text{err}_A = \frac{1}{N_t + 1} \sum_{j=0}^{N_t} \left| \theta_j^A - A(t_j) \right| \,,$$

$$\text{err}_B = \frac{1}{N_t + 1} \sum_{j=0}^{N_t} \left| \theta_j^B - B(t_j) \right| \,,$$

where $A(\cdot)$ and $B(\cdot)$ denote the true solutions. Here, we do not present the error of $\theta^C$ for two reasons. First, according to (45) and (49), the velocity field does not depend on $\theta^C$. Second, we observe from (47) and (51) that a constant shift of $\theta^C$ does not affect the regularization loss. Looking more carefully at (46), a constant shift of all the even (or odd) indices of $\theta^C$ also does not change the loss (47) or (51). Therefore, it is not meaningful to compute the error for $\theta^C$.

### B.4 Hyperparameters for the numerical examples

We present all the hyperparameters of the numerical tests in this section.

We set the diffusion coefficient $\gamma = 1$ for all problems, except for the double well potential example, where $\gamma = 0.1$. The reason for this difference is because if we set $\gamma = 1$ in the double well example, the terminal distribution $\rho(1, \cdot)$ will be very flat and we will not have a clear bifurcation pattern. In the LQ example in Appendix C.3, we set $\beta = 0.1$.

For all the numerical tests, we use a step size $\Delta t = 0.01$. The terminal time $t_{\text{end}} = 1$ for all the tests except for the double moon example. In the double moon example, we have a multi-stage splicing algorithm, with the first stage being $t \in [0, 0.2]$ and the second stage being $t \in [0.2, 0.4]$.

For the neural network parametrization (5), we use $\tanh$ as the activation function. The widths of the network are 20, 50, and 200 in 1, 2, and 10 dimensions for the RWPO and LQ problems. For the double moon flow matching problem (in 2 dimensions), the width is 200. For the double well example, the widths are 50, 100, and 200 in 1, 2, and 10 dimensions. The double moon flow matching problem and the double well problem are harder due to the bifurcation pattern.

For the regularized algorithms (Algorithm 2 and 3), we set the weight parameter for the HJB loss as $\lambda = 1 \times 10^{-3}$.

For all the examples, we train the model using the built-in Adam optimizer in Pytorch, with a learning rate 0.01. The number of training steps is set such that the training loss is roughly stable. The number of training steps are 200, 200, and 300 in 1, 2, and 10 dimensions for both RWPO and LQ problems with Algorithm 1, regularized WPO with Algorithm 2, flow matching for OU process with Algorithm 3. The number of steps for flow matching double moon example and the double well example in 1, 2, and 10 dimensions, the number of training steps is 500.

## C Additional numerical results

### C.1 RWPO

For the RWPO problem, after the transformation to probability flow through (16), the problem becomes

$$\inf_f \int_0^1 \int_{\mathbb{R}^d} \frac{1}{2} \left| f(t, x) + \gamma \nabla_x \log(\rho(t, x)) \right|^2 \rho(t, x) \, \mathrm{d}x \, \mathrm{d}t + \int_{\mathbb{R}^d} G(x) \rho(1, x) \, \mathrm{d}x \,,$$

subject to the transport equation

$$\partial_t \rho(t, x) + \nabla_x \cdot (\rho(t, x) f(t, x)) = 0 \,.$$

The optimal density evolution is given by

$$\rho(t, x) = (4\pi(2-t)\gamma)^{-\frac{d}{2}} \exp\left( -\frac{|x|^2}{4\gamma(2-t)} \right) \,,$$

and the optimal velocity field is

$$f(t, x) = -\frac{x}{2(2-t)} \,.$$

In addition to the errors and Figure 1, we present additional numerical results in Figure 2. The first, second, and third rows of Figure 2 show numerical results in 1, 2, and 10 dimensions, respectively. The first column shows the curves of the cost function through training. The light blue shadows plot the standard deviation observed during 10 independent test runs. The red dashed lines represent the cost under the optimal control field. We observe that our algorithm nearly reaches the optimal cost.

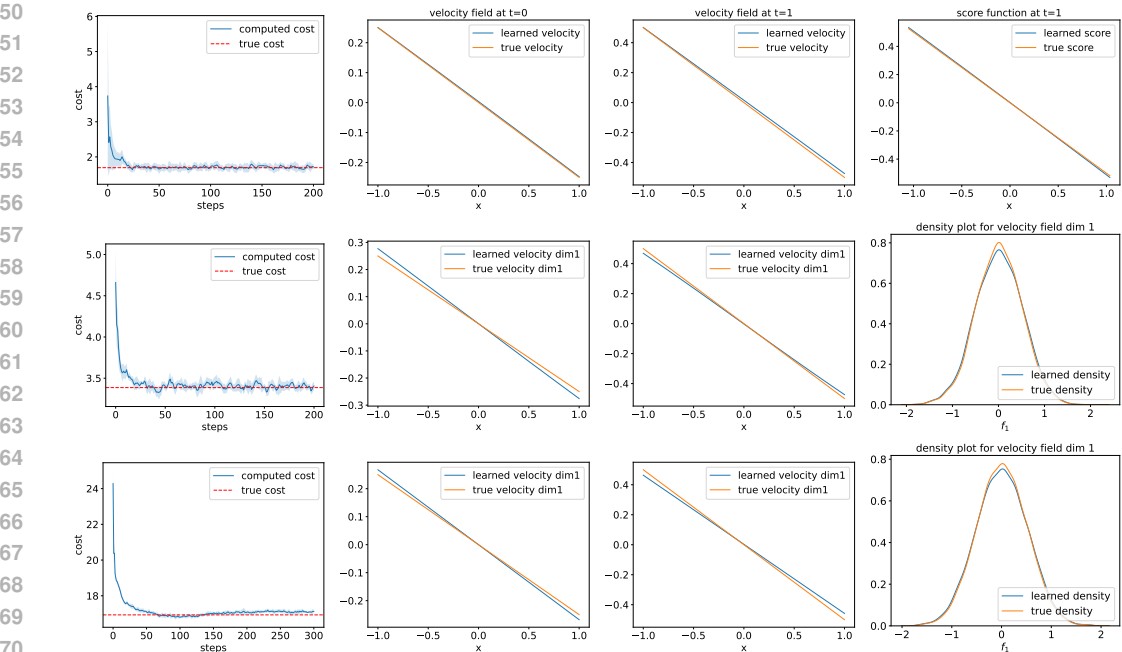

Figure 2: The first, second and third row: numerical results for RWPO in $1d$, $2d$, and $10d$. First column: cost functional through training. Second and third columns: first dimension of the velocity field at $t = 0$ and $t = 1$. Fourth column: the first row is the score function at $t = 1$, the second and third rows are the density plots for the first dimension of the velocity field. Our numerical results accurately capture the true solutions.

The second column in Figure 2 plots the first dimension of the velocity field at $t = 0$. Our neural networks (in blue) accurately captures the true velocity (in orange). The third column presents similar plots for the velocity fields, but at $t = t_{\text{end}} = 1$. Our algorithm also computes the velocity fields accurately.

The fourth column in the first row of Figure 2 plots the score function (in blue) at $t = 1$ using the data points $\{(z_{t_{N_t}}^{(n)}, s_{t_{N_t}}^{(n)})\}_{n=1}^{N_z}$. This curve coincides with the true score function in orange. Note that the score function at $t = 0$ is given, so we only plot the score at a terminal time. The second and third figures in the fourth column of Figure 2 show a density plot of the first dimension of the velocity field. These are probability density functions of $f(0, z_0; \theta)$ and $f(0, z_0)$, where $z_0 \sim \rho_0$. The density functions are further approximated by the histogram of samples. Such techniques for visualization of high dimensional functions have been applied by Han et al. (2020). We observe that our neural network accurately captures the velocity fields.

## C.2 Flow matching

In this section, we first present additional numerical results for the flow matching of the OU process. Then, we describe the details of the numerical implementation, such as the double moon potential, and present the numerical results.

**Flow matching for OU process.** The numerical results of flow matching for OU process are shown in Figure 3 and 4. Figure 3 shows the density evolution of the process in 1 dimension, which is similar to the second plot in Figure 1. Our algorithm captures both the change of mean value and the shrink of variance accurately. Figure 4 shows the particle dynamic under the trained probability flow in 2 dimensions and its comparison with the stochastic OU process. We add level sets of the density function with center $\mu(t)$ and radius $\frac{1}{4}\sigma(t)$, $\frac{1}{2}\sigma(t)$, and $\sigma(t)$ for better visualization, where $\sigma(t) := \det(\Sigma(t))^{\frac{1}{2d}}$ denotes the standard deviation. We pick points that are initialized within the circle with center $\mu(0)$ and radius $\sigma(0)$, and record the trajectories of these particles. According to Figure 4, our algorithm accurately captures the change of mean and variance of the OU process.

Additionally, compared with the OU process, our probability flow ODE demonstrates structured behaviors.

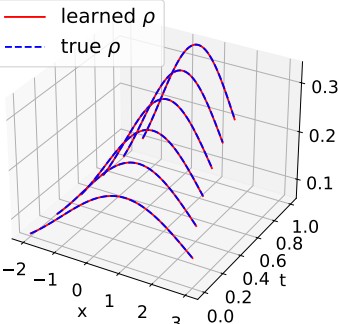

Figure 3: Flow matching for OU process in 1 dimension. Our algorithm accurately captures the density evolution of the state dynamic, including the change of mean and variance.

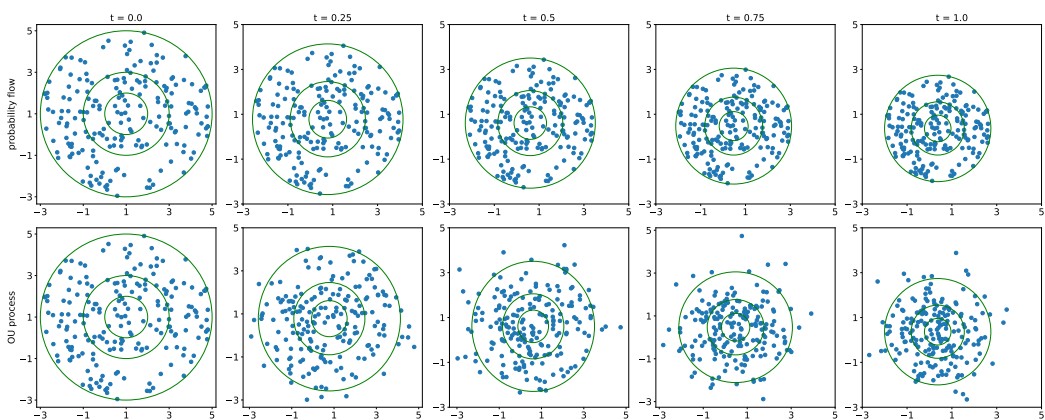

Figure 4: Flow matching for OU process in 2 dimension. First line: particle evolution under the probability flow $z_t$ with trained velocity field. Second line: simulation of the stochastic OU process. The circles in the figure are centered at $\mu(t)$, with radius $\frac{1}{4}\sigma(t)$, $\frac{1}{2}\sigma(t)$, and $\sigma(t)$. Our algorithm accurately captures the density evolution of the state dynamic, including the change of mean and variance. The probability flow demonstrate more structured behavior compared with the stochastic OU process.

**Double moon example.** In the example with double moon potential, given by

$$V(x) = 2(|x| - 3)^2 - 2\log\left[\exp\left(-2\left(x_1 - 3\right)^2\right) + \exp\left(-2\left(x_1 + 3\right)^2\right)\right]. \tag{54}$$

This example has been computed by Wang & Li (2022); Tan et al. (2023). We aim to learn the probability flow ODE of the overdamped Langevin dynamic

$$\partial_t \rho(t, x) - \nabla_x \cdot (\rho(t, x)\nabla_x V(t, x)) = \gamma\Delta_x\rho(t, x),$$

where the initial distribution $\rho_0$ is $N(0, 1)$. There are two moon-shaped patterns from this potential function. Therefore, the state dynamic has a bifurcation phenomenon, which is usually hard to capture.

We set the total time span as $t \in [0, 0.4]$. The overdamped Langevin dynamic is already close to its stable distribution at $t = 0.4$; see the last scattered plot in Figure 5. Also, our algorithm has demonstrated its ability to learn the dynamic accurately within a longer interval in other examples.

Next, we present the detailed implementation of the multi-stage splicing method for our double moon example. In this toy example, we partition the interval into two stages (sub-intervals) $[0, 0.2]$

and $[0.2, 0.4]$. We apply Algorithm 1 within the first interval. After training, we save the trained network as a warm-up (initialization) for the training in the next stage. We also record the particles $z_t^{(n)}$ at terminal time $t = 0.2$ in the first stage, which serves as the initial distribution for training in the next stage.

In the second stage $t \in [0.2, 0.4]$, the training process is similar. We inherit the network parameter $\theta$ from the previous stage as initialization. Also, the state particles $z_t^{(n)}$ at terminal time from the last stage are utilized as initialization distribution for the new stage. Note that starting form the second stage, there is no longer resampling because we only have finite samples at $t = 0.2$. As a consequence, we may encounter the problem of overfitting, which is also known as model collapse. In this work, we apply a $L^2$ regularization with weight $0.1$ to avoid this issue. If we only have finite samples for the initial distribution, then regularization should also be added in the first stage of training. We summarize this multi-stage splicing method in Algorithm 4.

The numerical result for the splicing method is shown in Figure 5. The first row shows the particle dynamic of state $z_t$ under the trained velocity field within the first stage, which coincides with an overdamped Langevin dynamic in the second row. The third row shows the particle dynamic in the second stage after training, coinciding with the overdamped Langevin dynamic in the fourth row. We also add the level sets for the density function of the stationary distribution for better visualization. These results confirm that our multistage splicing method can capture the Fokker-Planck equation of an overdamped Langevin dynamic in a total time span.

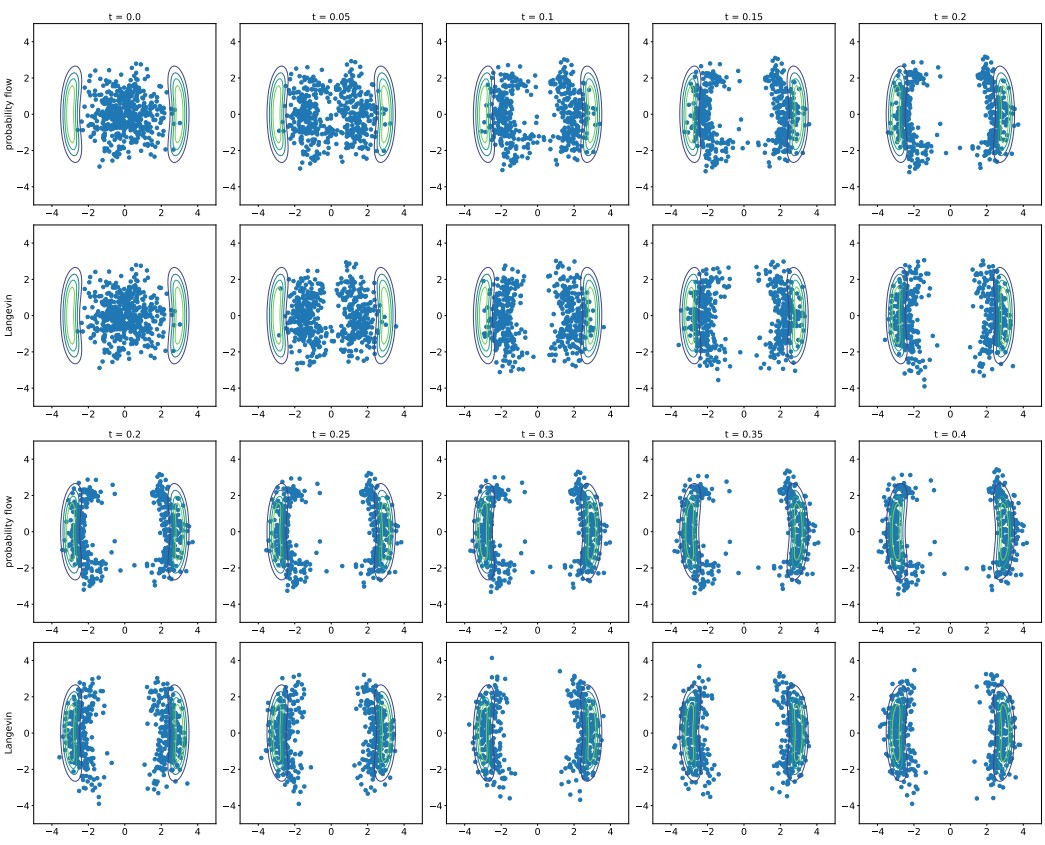

Figure 5: 2D flow matching double moon. First row: evolution of particles under the trained probability flow $z_t$ in the first stage $[0, 0.2]$. Second row: evolution of particles under the overdamped Langevin dynamic in the first stage $[0, 0.2]$. Third row: evolution of particles under the trained probability flow $z_t$ in the second stage $[0.2, 0.4]$. Fourth row: evolution of particles under the overdamped Langevin dynamic in the second stage $[0.2, 0.4]$. Our multi-stage splicing method captures the overdamped Langevin dynamic correctly.

### C.3 AN LQ EXAMPLE WITH AN ENTROPY POTENTIAL COST

We consider an LQ example in this section. This example was also studied by Lin et al. (2021). We minimize the cost functional

$$\inf_v \int_0^{t_{\text{end}}} \int_{\mathbb{R}^d} \left( \frac{1}{2} |v(t,x)|^2 + \frac{1}{2}|x|^2 + \beta \log(\rho(t,x)) \right) \rho(t,x) \, \mathrm{d}x \, \mathrm{d}t + \int_{\mathbb{R}^d} G(x)\rho(t_{\text{end}},x) \, \mathrm{d}x \,,$$

subject to

$$\partial_t \rho(t,x) + \nabla_x \cdot (\rho(t,x)v(t,x)) = \Delta_x \rho(t,x) \,, \qquad \rho(0,x) = \rho_0(x) \,.$$

With a score substitution, the problem is equivalent to

$$\inf_f \int_0^{t_{\text{end}}} \int_{\mathbb{R}^d} \left( \frac{1}{2} |f(t,x) + \nabla_x \log(\rho(t,x))|^2 + \frac{1}{2}|x|^2 + \beta \log(\rho(t,x)) \right) \rho(t,x) \, \mathrm{d}x \, \mathrm{d}t$$

$$+ \int_{\mathbb{R}^d} G(x)\rho(t_{\text{end}},x) \, \mathrm{d}x \,,$$

subject to

$$\partial_t \rho(t,x) + \nabla_x \cdot (\rho(t,x)f(t,x)) = 0 \,, \qquad \rho(0,x) = \rho_0(x) \,.$$

We define $\alpha := \left( \sqrt{\beta^2 + 4} - \beta \right)/2$. The initial distribution $\rho_0$ is Gaussian $N(0, \frac{1}{\alpha}I_d)$ and the terminal cost is $G(x) = \frac{\alpha}{2}|x|^2$. The optimal density evolution is given by

$$\rho(t,x) = \left( \frac{\alpha}{2\pi} \right)^{d/2} \exp\left( -\frac{\alpha |x|^2}{2} \right) \,.$$

We set $\beta = 0.1$ and test Algorithm 1 on this example. We remark that we have a term $\beta \log \rho(t,x)$ in the running cost $F$. Therefore, it is better to compute $l_t = \log \rho(t, z_t)$ instead of $\tilde{l}_t = \rho(t, z_t)$ in algorithm 1. Similar to the WPO example, we test our algorithm in 1, 2, and 10 dimensions. The errors are summarized in Table 5, which is similar to the results in Table 2. Our algorithm is able to solve the MFC problem accurately.

The numerical results are also presented in Figure 6. The first, second, and third rows show the results in 1, 2, and 10 dimensions, respectively. The first column is the training curve for the cost functional, which is similar to the first column in Figure 2. Our algorithm gives the correct objective. The second column visualize the density function at terminal time $t_{\text{end}}$ using the data points $\{z_{t_{N_t}}^{(n)}, \tilde{l}_{t_{N_t}}^{(n)}\}_{n=1}^{N_z}$. In the 1 dimension (first row), we plot the density and compare it with true densities directly. In 2 dimensions, we interpolate the data points and plot the density function $x_1 \mapsto \rho(t_{\text{end}}, (x_1, 0))$ in the figure. In 10 dimensions, an interpolation is very hard to obtain, so we plot the density in term of the norm of $x$. Our algorithm captures the density function accurately. The third column visualize the score function at terminal time $t_{\text{end}}$ using the data points $\{(z_{t_{N_t}}^{(n)}, s_{t_{N_t}}^{(n)})\}_{n=1}^{N_z}$. Again, we plot the score function directly in 1 dimension. In 2 and 10 dimensions, we present a density plot for the first dimension of the score, which gives the probability density function of $\nabla_{z_1} \log \rho(t_{\text{end}}, z_{t_{\text{end}}})$ (similar to the density plot in Appendix C.1). We observe that our algorithm captures the true density and score functions accurately.

| errors | $\text{err}_\rho$ | $\text{err}_f$ | $\text{err}_s$ | cost gap |
|--------|------|------|------|------|
| $1d$ | $9.91 \times 10^{-3}$ | $2.61 \times 10^{-2}$ | $2.88 \times 10^{-2}$ | $1.07 \times 10^{-2}$ |
| $2d$ | $7.39 \times 10^{-3}$ | $3.74 \times 10^{-2}$ | $3.58 \times 10^{-2}$ | $1.97 \times 10^{-2}$ |
| $10d$ | $1.15 \times 10^{-3}$ | $2.49 \times 10^{-2}$ | $2.37 \times 10^{-2}$ | $1.64 \times 10^{-1}$ |

Table 5: Errors for LQ problem.

### C.4 DOUBLE WELL POTENTIAL

In this section we present an MFC example where the terminal cost is a potential function with the double well shape. The formulation is similar to the WPO, with the terminal cost given by

$$G(x) = c |x - c_1|^2 |x - c_2|^2 \,, \tag{55}$$

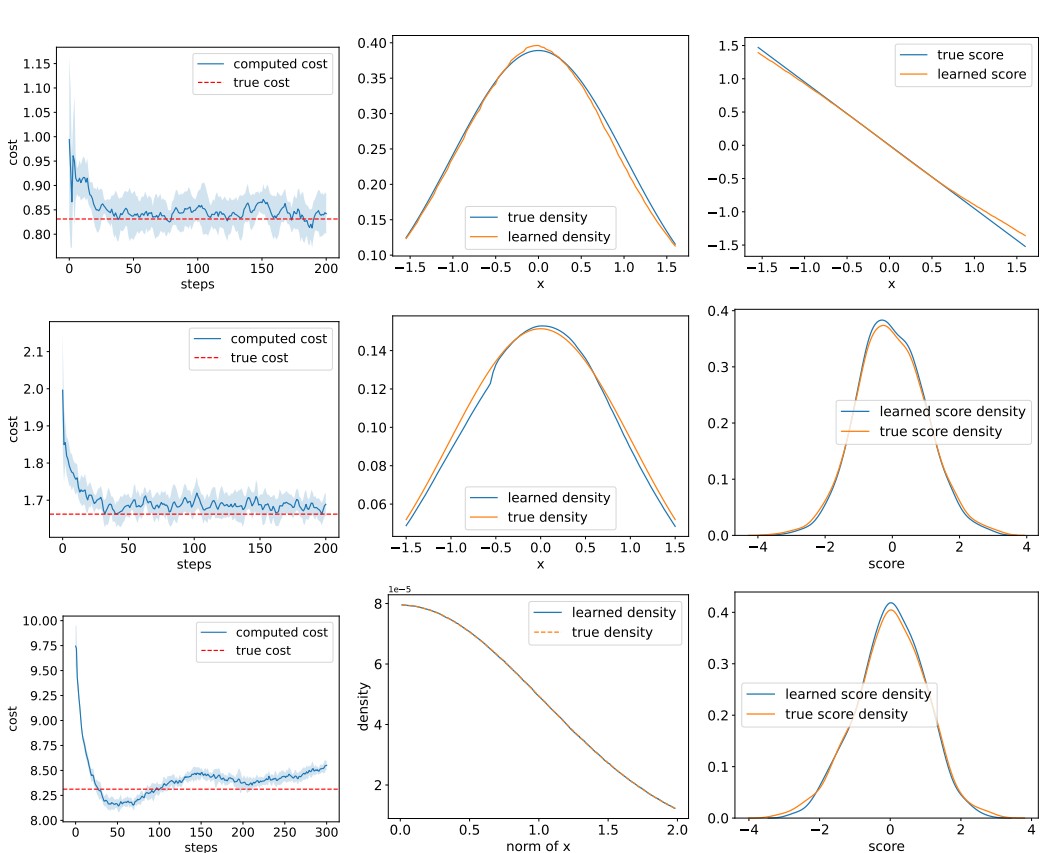

Figure 6: Numerical results for the LQ problem. The first, second, and third rows shows the results in 1, 2, and 10 dimensions respectively. First column: cost functional through training. Second column: visualization for the density function. Third column: visualization for the score function. Our algorithm accurately captures the solution to the problem.

where $c \in \mathbb{R}^+$ and $c_1, c_2 \in \mathbb{R}^d$ are the minimizers of $G$. This function $G$ does not depend on the density $\rho$. The initial distribution is still a standard Gaussian distribution, i.e., $z_0 \sim N(0, I_d)$. As mentioned before, such problem will demonstrate a bifurcation phenomenon, which is hard to capture. We compute this example in 1, 2, and 10 dimensions, and compute all errors in 1 and 2 dimensions. We take $c = \frac{1}{4}$ in (55) and set two centers as an all $-1$ vector for $c_1$ and an all 1 vector for $c_2$.

**Computing the reference solution.** Unlike the example in section 5.1, we do not have an explicit solution. But fortunately, we are able to obtain a reference solution using the kernel formula proposed by Li et al. (2023) (equation (10), (11), and (14)). According to their derivations, the optimal density function is given by

$$\rho(t, x) = \left(4\pi\gamma \frac{t(t_{\text{end}} - t)}{t_{\text{end}}}\right)^{-\frac{d}{2}} \int_{\mathbb{R}^d} \int_{\mathbb{R}^d} \frac{\exp\left[-\frac{1}{2\gamma}\left(G(z) + \frac{|x-z|^2}{2(t_{\text{end}} - t)} + \frac{|x-y|^2}{2t}\right)\right]}{\int_{\mathbb{R}^d} \exp\left[-\frac{1}{2\gamma}\left(G(\tilde{y}) + \frac{|y-\tilde{y}|^2}{2t_{\text{end}}}\right)\right] d\tilde{y}} \rho(0, y)\, dy\, dz\,.$$

(56)

Specifically, the density function as terminal time $t_{\text{end}}$ is

$$\rho(t_{\text{end}}, x) = \int_{\mathbb{R}^d} \frac{\exp\left[-\frac{1}{2\gamma}\left(G(x) + \frac{|x-y|^2}{2t_{\text{end}}}\right)\right]}{\int_{\mathbb{R}^d} \exp\left[-\frac{1}{2\gamma}\left(G(z) + \frac{|z-y|^2}{2t_{\text{end}}}\right)\right] dz} \rho(0, y)\, dy\,.$$

(57)

In addition, the solution to the classic HJB equation (cf. (39)) is given by

$$\phi(t, x) = 2\gamma \log\left(\int_{\mathbb{R}^d} (4\pi\gamma(t_{\text{end}} - t))^{-\frac{d}{2}} \exp\left[-\frac{1}{2\gamma}\left(G(y) + \frac{|x-y|^2}{2(t_{\text{end}} - t)}\right)\right] dy\right)\,,$$

with a terminal condition $\phi(T, x) = -G(x)$. With these expressions, we are able to obtain the score function via $\nabla_x \log \rho(t, x) = \nabla_x \rho(t, x)/\rho(t, x)$. Then, by Corollary 1 and (40), the optimal velocity is

$$f(t, x) = \nabla_x \phi(t, x) - \gamma \nabla_x \log \rho(t, x)\,.$$

(58)

However, when $t \in (0, t_{\text{end}})$, the numerical implementation for $\rho(t, x)$ and $\nabla_x \rho(t, x)$ through (56) involves three nested integrations in $\mathbb{R}^d$, which could potentially result in large errors. Therefore, we instead consider the errors at $t = 0$ and $t_{\text{end}}$, where all expressions are relatively easy. At $t = 0$, by (58), we have

$$f(0, x) = \frac{\int_{\mathbb{R}^d} \frac{x-y}{t_{\text{end}}} \exp\left[-\frac{1}{2\gamma}\left(G(y) + \frac{|x-y|^2}{2t_{\text{end}}}\right)\right] dy}{\int_{\mathbb{R}^d} \exp\left[-\frac{1}{2\gamma}\left(G(z) + \frac{|x-z|^2}{2t_{\text{end}}}\right)\right] dz} - \gamma \frac{\nabla_x \rho_0(x)}{\rho_0(x)}\,.$$

(59)

At $t = t_{\text{end}}$, we can compute the score function through

$$\nabla_x \log \rho(t_{\text{end}}, x) = \frac{-\int_{\mathbb{R}^d} \frac{-\frac{1}{2\gamma}\left(\nabla_x G(x) + \frac{x-y}{t_{\text{end}}}\right) \exp\left[-\frac{1}{2\gamma}\left(G(x) + \frac{|x-y|^2}{2t_{\text{end}}}\right)\right]}{\int_{\mathbb{R}^d} \exp\left[-\frac{1}{2\gamma}\left(G(z) + \frac{|z-y|^2}{2t_{\text{end}}}\right)\right] dz} \rho(0, y)\, dy}{\int_{\mathbb{R}^d} \frac{\exp\left[-\frac{1}{2\gamma}\left(G(x) + \frac{|x-y|^2}{2t_{\text{end}}}\right)\right]}{\int_{\mathbb{R}^d} \exp\left[-\frac{1}{2\gamma}\left(G(z) + \frac{|z-y|^2}{2t_{\text{end}}}\right)\right] dz} \rho(0, y)\, dy}\,.$$

(60)

Also, the terminal velocity is given by

$$f(t_{\text{end}}, x) = -\nabla_x G(x) - \gamma \nabla_x \log \rho(t_{\text{end}}, x)\,.$$

(61)

Next, we apply the Riemann sum to approximate the expressions (59), (60), and (61) to obtain a reference solution in 1 and 2 dimensions. First, we define the following function

$$h(y) = \int_{\mathbb{R}^d} \exp\left[-\frac{1}{2\gamma}\left(G(z) + \frac{|z-y|^2}{2t_{\text{end}}}\right)\right] dz\,,$$

which has appeared multiple times. We approximated $h(y)$ numerically from the Riemann sum. In 1 dimension, we use the trapezoid rule with step size $\Delta z = 0.01$ to approximate the integrations and truncate the integration within $z \in [-6, 6]$ when we compute the reference solution. In 2 dimensions,

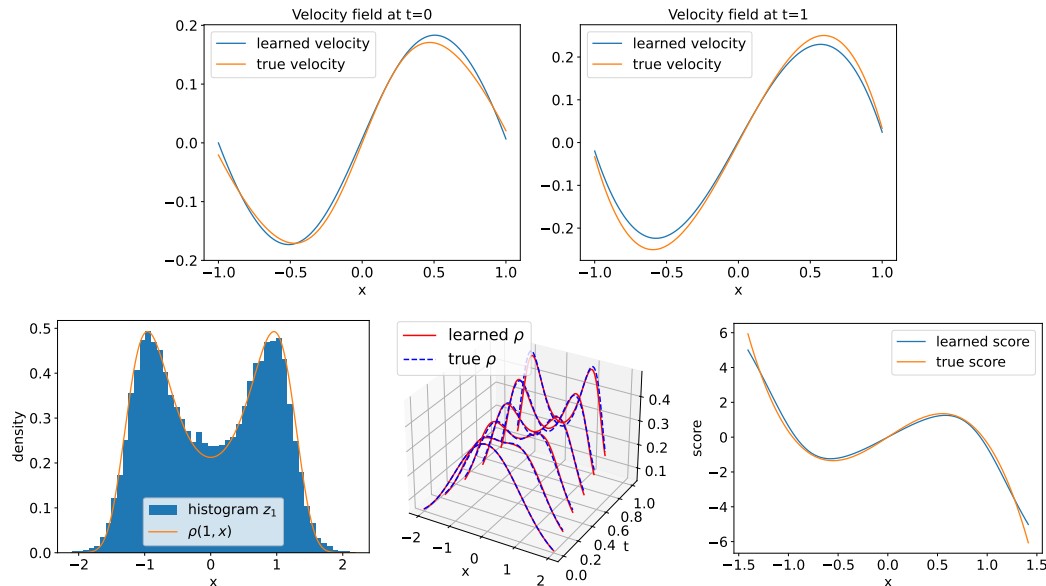

Figure 7: double well 1d. Left: true density and particle histogram at $T$; middle: density evolution compared with true density; right: score function at $T$.

similarly, we use a box with size $\Delta z_1 \times \Delta z_2 = 0.01 \times 0.01$ for Riemann sum and truncate the integration within the box $z \in [-6, 6] \times [-6, 6]$. We compute the value of $h(y)$ on all grid points within $[-4, 4]$ in 1 dimension and $[-4, 4] \times [-4, 4]$ in 2 dimensions, and store the values. In this way, integrations w.r.t. $z$ in (59) and (60) are simulated numerically. Next, we further apply Riemann sum for integrations w.r.t. $y$ with the same step size, truncated for $y \in [-4, 4]$ or $[-4, 4] \times [-4, 4]$. In this way, we obtain the function for all $x$ on the grid points. The two bounds 6 and 4, and the step size 0.01, are chosen such that the Riemann sums provide reasonable reference solutions for the problem. Finally, we make an interpolation in the dimensions 1 or 2, giving us the desired reference solution.

**Results in** 1 **dimension.** The errors for 1 dimensional example are $\text{err}_\rho = 1.17 \times 10^{-2}$, $\text{err}_{f_0} = 8.56 \times 10^{-3}$, $\text{err}_{f_{\text{end}}} = 2.39 \times 10^{-2}$, and $\text{err}_s = 1.65 \times 10^{-1}$. The numerical results are illustrated in Figure 7. The two plots in the first row show the velocity fields at $t = 0$ and $t = t_{\text{end}} = 1$, which nicely approximate the true velocities. In the second row, the figure on the left compares the reference terminal density and the histogram of particles $z_{t_{\text{end}}}$, which match well. The plot in the middle shows the evolution of the density, which is obtained using data points $\{(z_{t_j}^{(n)}, \tilde{l}_{t_j}^{(n)})\}_{j=0,n=1}^{N_t, N_z}$. These red lines coincide with the blue dashed lines, which are the true densities. This result demonstrates that our algorithm is able to capture the evaluation from a simple distribution to a complicated double well distribution accurately. The figure on the right plots the score function using data points $\{(z_{t_{N_t}}^{(n)}, \tilde{l}_{t_{N_t}}^{(n)})\}_{n=1}^{N_z}$. It successfully captures the S-shape of the true score function.

**Results in** 2 **dimensions.** It is hard to compute the nested integrations in Li et al. (2023)'s paper using Riemann sums in 2 dimensions. Therefore, instead of computing $\text{err}_f$ in (52), we compute the errors for the velocity field at $t = 0$ and $t = t_{\text{end}} = 1$ using the expressions in (53). The errors are $\text{err}_\rho = 2.62 \times 10^{-2}$, $\text{err}_{f_0} = 5.11 \times 10^{-2}$, $\text{err}_{f_{\text{end}}} = 4.77 \times 10^{-2}$, and $\text{err}_s = 2.4 \times 10^{-1}$.

The results are also shown in Figure 8 and 9. Our learned network successfully captures the velocity field $f$ and the score function which is shown in Figure 8. The first and second rows show each dimension of the velocity field at $t = 0$ and $t = t_{\text{end}} = 1$ and compare them with the true values. The third row shows the score function. The approximated score function is plotted using the data $\{(z_{t_{N_t}}^{(n)}, s_{t_{N_t}}^{(n)})\}_{n=1}^{N_z}$, where we made a non-uniform $2d$ interpolation using the build-in function in Scipy. The reference solution is obtained through differentiation of equations (10) (11) and (14) given by Li et al. (2023) and then apply Riemann sum approximation. We observe that our algorithm correctly captures the velocity field and the score function.

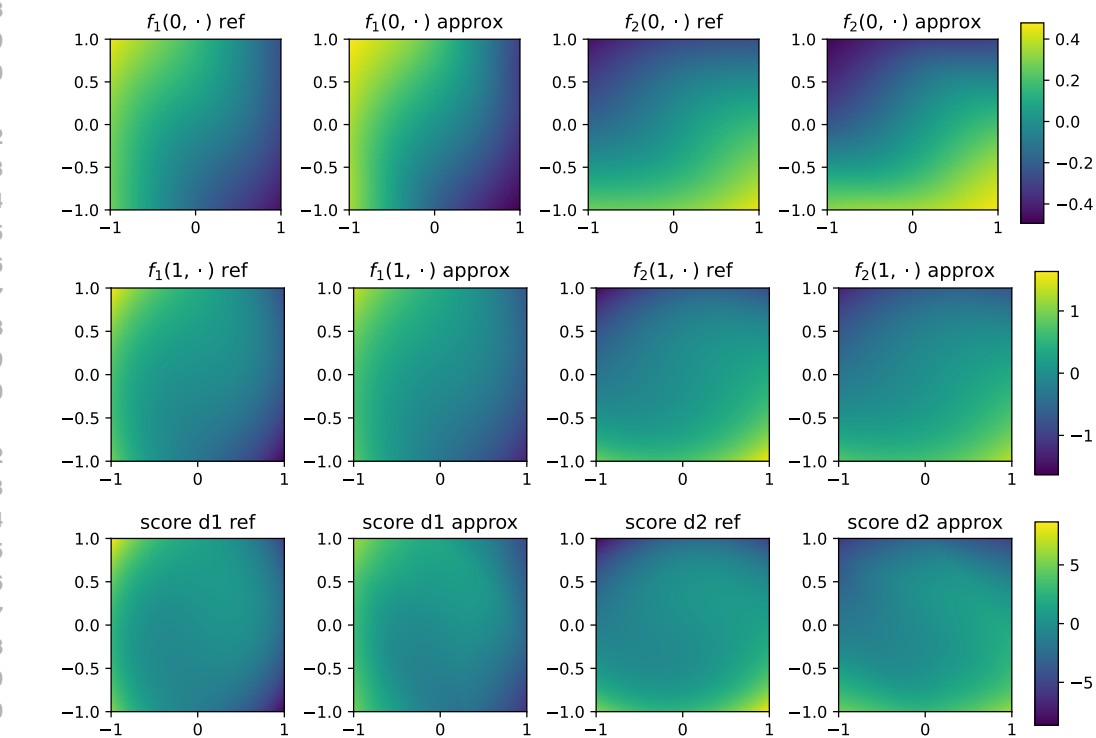

Figure 8: Image plot for $f(0, \cdot)$ (first row), $f(t_{\text{end}}, \cdot)$ (second row), and score function $\nabla_x \log \rho(t_{\text{end}}, \cdot)$. Columns one to four represent the true value in the first dimension, approximated value in the first dimension, true value in the second dimension, and the approximated value in the second dimension. Our algorithm correctly captures the velocity field and score function.

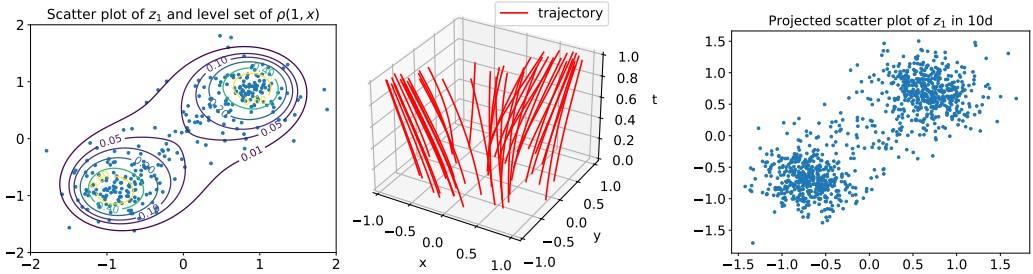

Figure 9: Left: scattered plot for particles at $t = t_{\text{end}}$ and the level sets of the density for 2 dimensional example. Middle: particle trajectory plots for 2 dimensional example, which demonstrates a bifurcation phenomenon. Right: projected scattered plot of particles at $t = t_{\text{end}}$ for 10 dimensional example.

The left plot in Figure 9 shows the scattered points at terminal time $t = 1$, which concentrate at the two wells of the potential. We also add the level sets of $\rho(1, \cdot)$ at $\{0.01, 0.05, 0.1, 0.2, 0.4, 0.6\}$ for better visualization. Note that the expression for $\rho(1, \cdot)$ is given by (57) instead of $\frac{1}{Z} \exp(-G(x)/\gamma)$. The plot in the middle shows the particle trajectories under the trained velocity field, which shows a bifurcation phenomenon.

**Results in 10 dimensions.**

Finally, we also test our algorithm in 10 dimensions. We do not have a reference solution in this case, but we observe that the particles go into two piles, as is shown in the right plot of Figure 9. In this plot, we project the 10 dimensional particles into 2 dimensions for visualization.

# D    REMARKS

We make a few remarks in this section.

**The score dynamic** (3c). Boffi & Vanden-Eijnden (2023) proposed a similar equation (in equation (13)) before. But we have concrete numerical example. Also, we believe that we are the first to propose the ODE dynamic for the second-order score function (3d). This equation significantly enhances the accuracy of the regularized MFC solver, as demonstrated in Table 3 and 4.

**Parametrization of neural network.** In this paper, we parametrize $f$ as a neural network with a single (hidden) layer. As a consequence, the whole map from $z_0$ to $z_{t_{\text{end}}}$ is slightly different from the typical structure of a deep neural network, as described by Chen et al. (2018). The layer structure for $f$ is affine-activation-affine, so the composition of multiple layers becomes affine-activation-affine-affine-activation-affine. In contrast, a typical structure for a deep neural network is affine-activation-affine-activation-affine. The composition of two affine functions is still an affine function, so it is not a big difference.

**Related work on theoretical MFC.** From a theoretical perspective, convergence properties of MFC problems have been extensively studied by Carmona & Laurière (2021; 2022). In addition, Lacker (2017) investigated the limit behavior as the number of agents approaches infinity. The maximum principle, a crucial concept in optimal control, has also been extended to the mean field setting by Meyer-Brandis et al. (2012), offering an understanding of the theoretical underpinnings of MFC problems. The mean field game (MFG) problem (Cardaliaguet, 2010) is closely related to MFC. Cardaliaguet et al. (2019) generalize the HJB equation to the master equation, whose properties have been further investigated by Gangbo & Mészáros (2022); Gangbo et al. (2022). Bayraktar et al. (2024) study the convergence of particle system on Wasserstein space.

**HJB regularization for MFC solver.** For the HJB regularization technique in B.1, we used absolute value instead of a squared residual in (48) because it performs slightly better. In fact, both forms of regularization could significantly reduce errors.

As for the reason why it reduces errors, we recall that we minimize the discretized loss function in the original Algorithm 1. As a consequence, there is a trend of overfitting to the discretized loss. Adding a regularizer could resolve this overfitting issue. Such issues for discretizations are also discussed in Zhou & Lu (2024).

**Computational cost for the score function via** (12) **and** (13). When we compute the score function through (12), the term $\nabla_x \log (\det(\nabla_x T(t, x)))$ could be tricky. Note that $\nabla_x \log (\det(\nabla_x T(t, x))) = \nabla_x \det(\nabla_x T(t, x)) / \det(\nabla_x T(t, x))$, and the $i$-th component of $\nabla_x \det(\nabla_x T(t, x))$ can be computed through

$$\partial_{x_i} \det(\nabla_x T(t, x)) = \det(\nabla_x T(t, x)) \operatorname{Tr}(\nabla_x T(t, x)^{-1} \partial_{x_i} \nabla_x T(t, x)).$$

Therefore, the $i$-th component of $\nabla_x \log (\det(\nabla_x T(t, x)))$ in (12) is

$$\operatorname{Tr}(\nabla_x T(t, x)^{-1} \partial_{x_i} \nabla_x T(t, x)).$$

As a consequence, the cost for computing one single score function $s_t$ through (12) is either from computing the inverse of Jacobi matrix or solving the related linear system for each dimension. Classical numerical linear algebra methods suggest that the complexity is $\mathcal{O}(d^3)$, and the cost for one score trajectory $\{s_{t_j}\}_{j=0}^{N_t}$ is $\mathcal{O}(N_t d^3)$.

We can also compute the score function through (13), where we assume that the width of the neural network is $\mathcal{O}(d)$, i.e., proportional the dimension. In this way, $\dfrac{\partial z_t}{\partial z_0}$ and $\dfrac{\partial l_t}{\partial z_0}$ are computed from auto-differentiations of the deep neural network functions in (11). The computational cost for a single score $s_{t_j}$ is $\mathcal{O}(j\, d^3)$, because we need to compute the back propagation of the deep neural network (11) with depth $j$ using chain rule. However, the total cost for computing a whole trajectory $\{s_{t_j}\}_{j=0}^{N_t}$ is $\mathcal{O}(N_t d^3)$ rather than $\mathcal{O}(N_t^2 d^3)$ because we can efficiently store intermediate computations. For example, according to (6), we can compute

$$\frac{\partial z_{t_{j+1}}}{\partial z_0} = \frac{\partial z_{t_{j+1}}}{\partial z_{t_j}} \frac{\partial z_{t_j}}{\partial z_0} = \left(I_d + \Delta t\, \nabla_z f(t_j, z_{t_j}; \theta)\right) \frac{\partial z_{t_j}}{\partial z_0}$$

with additional $\mathcal{O}(d^2)$ operations if $\dfrac{\partial z_{t_j}}{\partial z_0}$ is stored. Similarly, by (7), we can compute

$$\frac{\partial l_{t_{j+1}}}{\partial z_0} = \frac{\partial l_{t_j}}{\partial z_0} - \Delta t \left( \frac{\partial z_{t_j}}{\partial z_0} \right)^{\top} \nabla_z \left( \nabla_z \cdot f(t_j, z_{t_j}; \theta) \right)$$

with additional $\mathcal{O}(d^2)$ operations provided that $\dfrac{\partial l_{t_j}}{\partial z_0}$ and $\dfrac{\partial z_{t_j}}{\partial z_0}$ are stored. Note that computing the inverse of the Jacobian matrix or solving the related linear systems requires $\mathcal{O}(d^3)$ operations following classical numerical linear algebra methods. Therefore, computing $s_{t_{j+1}}$ through (13) requires additional $\mathcal{O}(d^3)$ operations if we smartly record the results at $t_j$. Consequently, the total cost for computing a whole trajectory $\{s_{t_j}\}_{j=0}^{N_t}$ is $\mathcal{O}(N_t \, d^3)$. We also remark that Huang et al. (2024) give a derivation similar to (13).

**Other methods to compute the score function.** For the **KDE** method, an approximation for the density function is obtained through taking the kernel convolution with samples directly. Then, one compute its spatial derivative and obtain the score function via $\nabla_x \log \rho(t, x) = \nabla_x \rho(t, x) / \rho(t, x)$. However, the choice of kernel and bandwidth could be tricky, and the numerical error could be large, especially in high dimensions (Węglarczyk, 2018). **Score matching** (Hyvärinen & Dayan, 2005) is another commonly used method. With a small trick of integration by part and dropping a constant term that includes the integral with squared score functions, one only requires samples to train a network for the score function. However, training a neural network could be expensive, compared with the KDE method, which evaluates the score function directly.

**Regularity for the MFC problem.** Formula (3) requires third order derivatives for the velocity field $f$. As a consequence, we need regularity assumption for the MFC problem to guarantee that the ODE system is meaningful. In this work, we assume that $L$ and $F$ are $C^{1,2,2}$ Hölder continuous, and $G$ is $C^{2,2}$ Hölder continuous. According to Schauder estimates (Ladyzhenskaia et al., 1968, Chapter 4), these regularity assumptions guarantee that the velocity field has up to third-order derivatives, ensuring that the ODE system (3) is meaningful.

**The multi-stage splicing method.** For our multi-stage splicing method (cf. Appendix C.2), we partition the total time span evenly and pass the trained neural network from the previous stage into the next stage. When the dynamic becomes complicated, the length of the time interval could also be adapted to suit the problem (Zhou et al., 2021).

Also, the neural network from the previous stage is probably not an accurate velocity for the new stage, but it does contain more information than a random initialization. Therefore, this nice initialization could serve as a warm-up (Zhou et al., 2023) for the training in the new stage, which could accelerate the convergence of this algorithm.

**Different numerical schemes for time discretizations.** In the numerical tests, we have also tried other schemes besides the forward Euler scheme, such as the implicit Euler scheme, the mid-point scheme, and the Runge-Kutta scheme. The numerical results for different schemes are similar. We believe that the time discretization error is not the bottleneck of our algorithm at this moment. It is probably more important to train the neural network properly.

