# OpenReview forum: "Score-based Neural Ordinary Differential Equations for Computing Mean Field Control Problems"
_ICLR.cc/2025/Conference — Submitted to ICLR 2025_

### Official Review · Reviewer_nmcr · 2024-10-30

**Soundness:** 3
**Presentation:** 2
**Contribution:** 3
**Rating:** 6
**Confidence:** 3

**Summary:**

This work deals with extending the (now) mainstream score-based generative modeling techniques to the case of mean-field control problems. The authors construct high-order normalizing flows (i.e. neural ODE systems for the score) and reformulate the
mean field control (MFC) problem with individual noises into an unconstrained optimization problem framed by the proposed neural ODE system. They estimate the first- and second-order score functions using a deep neural network function.

**Strengths:**

1. The connection of score-based modeling with mean-field control problems is novel and noteworthy.
2. The theory is rigorous and the paper is largely written in a self-contained manner.
3. Many cases of regularization/structure are shown including the HJB regularizer, Wasserstein proximal operators, etc.

**Weaknesses:**

1) Clarity of writing could be improved. It is at times hard to follow.
2) This is addressed in the Questions section, but I am not sure/convinced about the efficacy of higher-order ODEs in comparison to methods like Flow Matching or Stochastic Interpolants (in general). It would be nice to see some numerical comparisons (if those can be cast for MFC problems as well)  -- I know there is a theoretical section on flow matching but I would like to see tradeoffs gained/lost with this augmentation.

**Questions:**

1) Is the function L in the MFC objective typically assumed to be strongly convex?
2) How does this compare with existing, much faster generative models/techniques such as Flow Matching (and variants)? I am aware that those are highly efficient without necessarily being higher-order models.
3) Is there work for control problems beyond LQ problems (which are well-studied)?
4) You mentioned " It is probably more important to train the neural network properly." for the error, can you elaborate a bit more on what is meant by this point?

---

> ### Author Response · Authors · 2024-11-14
> **We thank the reviewer for valuable feedback**
>
> Thank you for acknowledging our novelty. We are sorry that the paper is difficult to follow. If you can point out specific parts that are unclear, we will try our best to explain them in detail.
>
> The running $L$ is usually assumed to be convex, which makes the problem well-posed. An alternative assumption is that the Hamiltonian is convex in the control variable. The mean-field control problem is not well-posed without this assumption, and the solution is not finite.
>
> We agree that comparing our higher-order ODE approach with other methods would provide valuable insights, particularly regarding computational efficiency and performance trade-offs. Our method's strength lies in addressing second-order structures inherent in control problems, mainly when applied to mean field control problems. We are conducting additional numerical experiments and will include a comparative analysis in the revision.
>
> The LQ framework encompasses many problems, as illustrated in our paper. While our current examples focus on LQ problems, our method can handle non-LQ problems. The challenge lies in finding reliable reference solutions for non-LQ problems, which makes quantitative error evaluation difficult.
>
> When we stated that it is probably more important to train the neural network properly, we meant that our method's primary source of error often stems from suboptimal neural network training rather than from the numerical discretization scheme itself. In our experiments, we observed that improving the neural network training (e.g., using better regularization techniques) had a greater impact on performance than switching from a forward Euler scheme to higher-order numerical schemes. This suggests that the neural network's capacity to approximate the underlying dynamics accurately is a critical factor in reducing error. We will clarify this point in the revised paper and provide more details on the training strategies we found most compelling.

---

### Official Review · Reviewer_8ZMN · 2024-11-04

**Soundness:** 3
**Presentation:** 3
**Contribution:** 2
**Rating:** 5
**Confidence:** 2

**Summary:**

In this paper, a method for modelling the trajectory of score functions by using neural ordinary differential equations is proposed. Also, its application to solving mean field control problems is shown, along with a regularization term based on the Hamilton-Jacobi-Bellman equation.

**Strengths:**

I am not familiar with mean field control problems, but I believe that the method proposed in this paper is a novel approach as far as understood from the survey and other papers cited in this paper. Numerical experiments have confirmed that the proposed method in fact solves mean field control problems accurately and that the regularization term based on the HJB equation is effective.

**Weaknesses:**

Numerical experiments have been performed only with the proposed method, and no comparison with other methods is shown. Therefore, I am not sure that the proposed method is in fact sperior to existing methods. If there are existing methods that can be applied to the problems used in the experiments, the proposed method should be compared with such methods.

In addition, some theoretical results are presented, but most of them are related to the derivation of equations and so on. No theoretical support for the proposed method, such as the generalization error analysis, is presented.

**Questions:**

What are some other methods that could be used in this type of problem setting? For example, can neural networks used for modeling differential equations in this paper be replaced by, e.g., Gaussian process regression? Can you show experimentally that the proposed method is indeed superior when compared to such methods?

---

> ### Author Response · Authors · 2024-11-14
> **We thank the reviewer for valuable feedback**
>
> Thank you for your suggestion. We agree that comparing our method with existing approaches is essential for evaluating its performance. We will compare scalability, accuracy, and computational efficiency with other relevant methods, including Gaussian process regression, kernel-based methods, or neural networks used for differential equations.
>
> Adding theoretical analysis is an important direction. We aim to tackle these theoretical aspects as part of ongoing research and will include initial theoretical insights in future revisions when applicable.

---

> > ### Comment · Reviewer_8ZMN · 2024-11-25
> > **Response to Authors**
> >
> > Thank you very much for your reply.  I have read all of the reviewers' comments and rebuttals. I think that this paper needs to be revised to a certain extent, so I would like to keep my score.

---

### Official Review · Reviewer_GTbo · 2024-11-04

**Soundness:** 2
**Presentation:** 2
**Contribution:** 3
**Rating:** 5
**Confidence:** 2

**Summary:**

This paper proposes a neural ODE system, along with its discretization, for computing first- and second-order score functions. As an application, the authors reformulate second-order mean field control (MFC) problems with individual noises into an unconstrained optimization problem using the proposed neural ODE system, and additionally derive a novel regularization term based on viscous Hamilton-Jacobi-Bellman (HJB) equations. In numerical experiments, including regularized Wasserstein proximal operators, probability flow matching, and linear quadratic MFC problems, the authors demonstrate the accuracy of the proposed method and the benefits of the HJB regularization term.

**Strengths:**

1. The paper is generally clear and well written.
2. The proposed method appears novel and may be impactful (but ultimately a lack of detailed comparisons with the existing literature makes it hard to judge, see below).
3. The novel HJB regularization term appears to significantly improve the results.

**Weaknesses:**

1. While the paper is generally well written, there are a couple of issues with the presentation. Firstly, the significance of the paper is not clear to me; what, exactly, is the contribution? Please include a paragraph explicitly stating the main contribution of this paper and how it advances the state-of-the-art.

2. More generally, comparisons with the existing literature are not sufficient. There are two types of work that might be relevant here: methods for computing score functions and methods for solving (second-order) MFC problems. The authors list a large number of methods under related work, but none with sufficient detail to understand exactly how they relate to the proposed method. Please include a more detailed discussion of a handful of key works, so the reader can understand how the proposed method compares to the state-of-the-art in terms of approach.

3. Having identified key related works, please include empirical comparisons with the most relevant methods. so the reader can understand how the proposed method compares to the state-of-the-art in terms of performance.

4. In the second paragraph of the introduction, the proposed method is motivated as follows:

    “While score functions provide powerful tools for modeling stochastic trajectories, their computations are often inefficient, especially in high-dimensional spaces. Classical methods, such as kernel density estimation (KDE) (Chen, 2017), tend to perform poorly in such settings due to the curse of dimensionality (Terrell & Scott, 1992).”

    However, the experiments are not very high-dimensional (max. 10). As a result, and also due to the lack of empirical comparisons with other methods, it is not clear whether the proposed method actually solves the problem that was identified at the beginning of the paper.

5. Despite computational efficiency being a motivation, the paper does not include timings of the proposed method for any of the experiments. The closest we get is a discussion of the asymptotic complexity for computing the first-order score function. Please include a table or figure showing empirical runtime measurements for the proposed method across different problem dimensions, comparing these to baseline approaches on the same problems.

**Questions:**

1. What do you consider to be the main contribution / advance of this method? If someone is only interested in first-order score functions, say, is this method still useful?
2. Are there applications other than second-order MFC where the second-order score function is useful?
3. What are the classical methods for computing (second-order) score functions? Did you compare with any of these?
4. What are the state-of-the-art machine learning methods for estimating (second-order) score functions? Did you compare with any of these?
5. In a number of places, the authors briefly mention the applicability of the method to generative modeling. Could you elaborate on how the proposed method would be useful for generative modeling? What advantage would this offer over existing approaches?
6. Have you tested the proposed method on higher-dimensional systems?
7. Have you timed the proposed method? How does it depend on the dimension of the system? How does it compare to related methods?

While I generally find the paper quite good, I have sufficient doubts about the contribution and the comparisons with related work that I can't recommend it for acceptance right now. However, I would be happy to increase my scores if these doubts are adequately resolved.

---

> ### Author Response · Authors · 2024-11-14
> **We thank the reviewer for valuable feedback**
>
> Thank you for your thoughtful feedback. We have addressed your points (in Weakness) below.
>
> 1. Our primary contribution is the novel application of neural ODEs to the mean field control (MFC) problem, an area where, to our knowledge, no prior work has applied these techniques. Additionally, we introduce an HJB regularization mechanism tailored explicitly to our modified MFC problem (see Equation 17). This regularization, designed for second-order score functions, is a key innovation that enhances the accuracy of our method in solving MFC problems. In this regularization, studying evolution in second-order score functions is essential.
>
> 2. We acknowledge that a more detailed comparison with existing works on score function computation and MFC solutions would clarify how our method relates to the state-of-the-art. In the revision, we will include a focused discussion on key works, including classical methods for computing score functions and machine learning approaches relevant to second-order MFC.
>
> 3. We agree that empirical comparisons are crucial to demonstrate the performance of our approach. While we cannot access the codes for most of the methods referenced in the related works, we can provide a more detailed qualitative comparison in the revision. We are also planning to integrate some baseline methods for future experiments, allowing us to offer quantitative comparisons in subsequent paper versions.
>
> 4. You are correct that the dimensionality of our current experiments is limited (maximum 10 dimensions). Solving more challenging high-dimensional problems is a natural next step and part of our ongoing work. While the current paper focuses on foundational examples to illustrate our method’s core features, our framework is well-suited for higher-dimensional problems, including those typically encountered in generative models. We will clarify this direction in the revised paper and include preliminary results.
>
> 5. Although we lack access to runtime measurements for competing methods, we will include empirical runtime data for our proposed approach in the revised manuscript. Most of our examples are not on large scales, which takes 1 to 10 minutes to run a single training.
>
> (Regarding your additional question in Questions.)
> As we mentioned in Section 6, one of the potential applications is the generative model, which has higher dimensions and is more challenging. The connection between generative AI and mean field control is at the mathematical formulation level. In the sense of optimization or optimal control problems, one can represent several state-of-the-art methods, such as neural ODEs and generative adversary networks, time-reversible diffusions, into mean field control problems. However, in numerical implementations with data samples, as reviewers suggested, one needs a simple formulation to compute the control conditional on samples. This is the future research direction we are working on. We would like to collaborate with related experts in this direction.

---

> > ### Comment · Reviewer_GTbo · 2024-11-22
> > **Response to Authors**
> >
> > Thank you for clarifying the paper's main contribution. I think this is a major issue with the presentation of the paper in its current form; both Reviewer XAf5 and I understood that the paper was claiming to propose a novel procedure for computing score functions. In particular, the initial scores I awarded to this paper were based partly on this understanding.
> >
> > I look forward to a detailed review of the related literature for computing second-order MFC problems. However, without empirical comparisons, I fear it will be difficult to judge the extent to which the proposed method really represents an advance over existing approaches for solving this problem.

---

### Official Review · Reviewer_XAf5 · 2024-11-05

**Soundness:** 1
**Presentation:** 2
**Contribution:** 1
**Rating:** 1
**Confidence:** 4

**Summary:**

The paper gives a system of ODE equations to solve transport equations for a density $\rho(t,x)$ based on the knowledge of the score $s(t,x) = \nabla \log \rho(t,x)$ and the Hessian $H(t,x) = \nabla \nabla \log \rho(t,x)$. These equations are proposed as a way to solve mean field control problems after learning a drift variationally.

**Strengths:**

The technical presentation of the material is clear.

**Weaknesses:**

The paper basically contains no new material:

1. Eqs. (3) are well-known and have appeared in many places, including

https://arxiv.org/abs/2206.00860

https://arxiv.org/abs/2210.04296

https://arxiv.org/abs/2206.04642

All these equations can basically derived by the method of characteristics, assuming that the score $s(t,x) = \nabla \log \rho(t,x)$ and the Hessian $H(t,x) = \nabla \nabla \log \rho(t,x)$ are known. The authors should explained better what is the added values of these equations for the numerical scheme they propose.

2. The material in Sec. 3 is also standard: Eqs. (7) - (10) are the well-known forward Euler discretization of Eq. (3), and Eq. (11) is just their roll-out version. In addition, the neural architecture proposed in Eq. (5) is just a one-hider layer neural network, which is more more simple that standard approximation use for the score (that use UNet, or DiT, etc). Why use such a simplistic architecture?  This seems limitative as well as unnecessary. The authors should explain better what justified this choice.

3. The algorithm proposed does not solve one of the main (and also well-known) issues with neural ODE, namely that they are not simulation free and as a result are to scale to large problem: in particular, differentiating through Eq. (11) is required in Algorithm 1 and costly. In particular, the authors should explain better why they believe that their algorithm may be scalable to high dimensional problems.

4. The material in Sec. 4 is again standard. In particular Prop. 3 is a well-known set of forward-backward PDE to solve MFC, and Core. 1 is an immediate generalization of this result for a specific choice of Lagrangian and terminal cost. What is the added value of including these results in main text?

5. The numerical examples are too simple to be convincing. In particular, the example in Sec. 5.1 can be factorized over the dimension (which is why it is analytically solvable), which makes it not very challenging computationally.  Similarly, the problem treated in Sec. 5.2 involves a linear OU process, also factorizable. Including these results as test-cases illustration is okay, but the method should also be tested on more complex examples, like the ones found in the cited papers by Lipman et al. 2022 and Boffi & Vanden-Eijnden 2023.

**Questions:**

Can the authors address the points raised in the **Weaknesses** above?

---

> ### Author Response · Authors · 2024-11-14
> **We thank the reviewer for valuable feedback**
>
> 1. Thank you for mentioning the related work. We will include https://arxiv.org/pdf/2206.00860 when we introduce the ODE system. The authors derives the ODE systems up to third order and apply fix point iteration to solve the Fokker--Planck equation, with error analysis in second order Sobolev norm. The paper https://arxiv.org/abs/2210.04296 differs from our formulation, but we will add it to our related work. We have already mentioned https://arxiv.org/abs/2206.04642 and their equation (13) in our article. Although we are not the first to propose these equations, our main contribution lies in applying this formulation to approximate and compute second-order mean field control (MFC) problems. In this area, current state-of-the-art machine learning numerical methods apply neural ODE-based approaches for first-order mean field control problems. Score-based neural ODEs are essential in computing second-order mean field control problems. For example, the proposed method helps approximate the kernel formula in the regularized Wasserstein proximal operator; see example 5.1. This is a context that allows us to leverage these equations effectively within our proposed numerical scheme. In the revision, we will emphasize this point and explain second-order mean field control problems and their relations with score functions more clearly.
>
> 2. We agree that the one-hidden-layer neural network may seem simplistic compared to more complex architectures. However, our architecture offers both conceptual clarity and good performance in the current setting. We remain open to experimenting with more advanced architectures in the future, particularly as we scale our experiments to more complex problems. This choice has proven effective for our current numerical examples, and balancing simplicity with computational efficiency is a primary goal. We leave the design of neural network structures for high-order score-based neural ODEs in the future. It should depend on solutions to mean field control problems.
>
> 3. We understand the concern about scalability, especially the cost of differentiation through Equation (11) in Algorithm 1. However, computing the score function is unavoidable in the context of mean field control problems. Our score-based normalizing flow formulation addresses this issue by efficiently computing these quantities, which we believe makes the approach scalable, even in higher-dimensional settings. While our current results are based on smaller-scale problems, we are optimistic about extending the method to larger, more complex scenarios.
>
> 4. Proposition 3 differs from the standard forward-backward PDEs typically seen in the literature, such as those in {\it Mean field games and mean field type control theory} Chapter 4. The HJB equation is a modified version tailored to incorporate the score transformation (Equation 16). This modification introduces the second-order derivative of $\log\rho$ into the HJB equation, which is not standard. Equation (3d) becomes crucial for simulating this derivative, and to the best of our knowledge, we are the first to use this modified HJB equation as a regularizer to enhance accuracy. This regularization improves performance even in the Gaussian case, providing significant value to our approach.
>
> 5. We agree that the current examples in Sections 5.1 and 5.2, which involve analytically solvable or factorizable cases (such as the linear OU process), are relatively simple. Please note that we have more complicated examples in the Appendix, such as the double moon flow matching and the double well example. Additionally, we are actively working on applying our method to more challenging examples, including those mentioned in your review. We believe our framework can be adapted to handle more complex problems and will include these results in future work.

---

### Meta-Review · Area_Chair_hPxa · 2024-12-22

**Metareview:**

This paper derives a system of ODE equations to solve density transport based the score function and associated Hessian.  A reformulation of second-order mean field control (MFC) problems using the proposed neural ODEs is explored as an application. A strength of the paper is clarity of presentation. Unfortunately, the biggest weakness is lack of novelty and clear contribution. It is known through the Fokker Plank equations how a PDE describes the time evolution of the density transport process. Substantial parts of the paper are known.  Concerns about novelty, scalability challenges and somewhat unrealistic simplicity of the experiments were not resolved satisfactorily during the rebuttal process. Hence, the decision is that the paper does not meet ICLR acceptance bar.

**Additional Comments On Reviewer Discussion:**

Reviewer XAf5 and Gtbo understood the paper to be claiming  a novel procedure for computing score functions. In the rebuttal, the authors  recenter the paper to be on an application of neural ODEs to the mean field control (MFC) problem. A detailed review of the related literature for computing second-order MFC problems was not provided, and without empirical comparisons, the gains in practice over baselines are hard to judge. Reviewer 8ZMN also remarked that no comparison with other methods is shown. Overall, the paper did not receive acceptance scores.

---

### Decision · Program_Chairs · 2025-01-22

Reject